# BLIP-Diffusion: Pre-trained Subject Representation for Controllable Text-to-Image Generation and Editing

**Dongxu Li[†], Junnan Li[†], Steven C.H. Hoi[†]**
Salesforce AI Research
[†]Corresponding authors: {li.d,junnan.li,shoi}@salesforce.com
https://github.com/salesforce/LAVIS/tree/main/projects/blip-diffusion

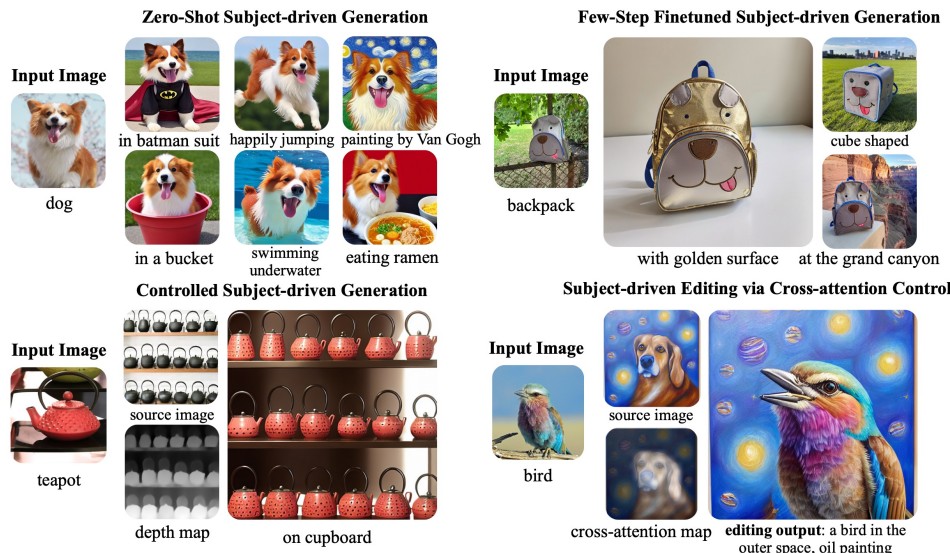

Figure 1: Leveraging the pre-trained subject representation, BLIP-Diffusion enables subject-driven generation under efficient fine-tuning or zero-shot setups. The model can also serve as a foundation subject-driven text-to-image generation model and supports applications such as controlled generation and image editing, when combined with techniques such as ControlNet [1] and prompt-to-prompt [2].

## Abstract

Subject-driven text-to-image generation models create novel renditions of an input subject based on text prompts. Existing models suffer from lengthy fine-tuning and difficulties preserving the subject fidelity. To overcome these limitations, we introduce BLIP-Diffusion, a new subject-driven image generation model that supports multimodal control which consumes inputs of subject images and text prompts. Unlike other subject-driven generation models, BLIP-Diffusion introduces a new multimodal encoder which is pre-trained to provide subject representation. We first pre-train the multimodal encoder following BLIP-2 to produce visual representation aligned with the text. Then we design a subject representation learning task which enables a diffusion model to leverage such visual representation and generates new subject renditions. Compared with previous methods such as DreamBooth, our model enables zero-shot subject-driven generation, and efficient fine-tuning for customized subject with up to 20x speedup. We also show that BLIP-Diffusion can be flexibly combined with existing techniques such as ControlNet and prompt-to-prompt to enable novel subject-driven generation and editing applications.

37th Conference on Neural Information Processing Systems (NeurIPS 2023).

# 1 Introduction

Text-to-image generation models have developed significantly and enabled creation of high-quality images based on textual prompts [3–7]. One of their applications is subject-driven generation, which aims to render novel renditions of an input subject while preserving its appearance. The common approach to subject-driven generation [8–11] is through inverting subject visuals into text embedding space. Specifically, with a pretrained text-to-image generation model, a placeholder text embedding is optimized to reconstruct a set of subject images. The embedding is then composed into natural language prompts to create different subject renditions. One known inefficiency of this approach is that it requires reiterating hundreds [9, 10] or thousands [8] tedious fine-tuning steps for each new subject, which hinders it from efficiently scaling to a wide range of subjects.

We attribute such inefficiency to the fact that most pre-trained text-to-image models do not natively support multimodal control - using both images and texts as control input. As a result, it becomes challenging to learn subject representation that aligns with the text space while capturing the subject visuals with high fidelity. To overcome these limitations, we introduce BLIP-Diffusion, the first subject-driven text-to-image generation model with *pre-trained generic subject representation*, which enables subject-driven generation in zero-shot or with few-step fine-tuning. Our model builds upon a vision-language encoder (*i.e.* BLIP-2 [12]) and a latent diffusion model [6] (*i.e.* Stable Diffusion). The BLIP-2 encoder takes as input the subject image and its category text; it produces text-aligned subject representation as output. We then infix the subject representation in the prompt embedding to guide the latent diffusion model for subject-driven image generation and editing.

To enable controllable and high-fidelity generation, we propose a new two-stage pre-training strategy to learn generic subject representation. In the first pre-training stage, we perform multimodal representation learning, which enforces BLIP-2 to produce text-aligned visual features based on the input image. In the second pre-training stage, we design a subject representation learning task where the diffusion model learns to generate novel subject renditions based on the input visual features. To achieve this, we curate pairs of input-target images with the same subject appearing in different contexts. Specifically, we synthesize input images by composing the subject with a random background. During pre-training, we feed the synthetic input image and the subject class label through BLIP-2 to obtain the multimodal embeddings as subject representation. The subject representation is then combined with a text prompt to guide the generation of the target image.

Benefiting from the pre-trained subject representation, BLIP-Diffusion achieves promising zero-shot subject-driven generation results and superior fine-tuning efficiency. For example, BLIP-Diffusion takes 40-120 fine-tuning steps to specialize for a given subject, achieving up to 20x speedup compared to DreamBooth [9]. Furthermore, BLIP-Diffusion inherits behaviours of the constituent latent diffusion model and can be flexibly extended to support various subject-driven generative applications without further training. Following the prompt-to-prompt [2] approach, BLIP-Diffusion enables editing images with subject-specific visuals. When combined with ControlNet [1], it enables subject-driven generation with various additional structure control.

# 2 Related Work

## 2.1 Diffusion Models for Text-to-Image Generation

Diffusion models [3, 4, 6, 7, 13–15] generate images by progressively denoising a random variable drawn from a Gaussian distribution. In this work, we are particularly interested in the pre-trained text-to-image latent diffusion models [6]. Given a latent variable $z$ and its noisy version $z_t$ obtained by gradually adding noises to $z$ for $t$ steps, latent diffusion models optimize the following objective:

$$\mathbb{E}_{z,c,\epsilon\sim\mathcal{N}(0,1),t}\left[\|\epsilon - \epsilon_\theta(z_t,t)\|_2^2\right], \tag{1}$$

which is the squared error between the added noise $\epsilon$ and the predicted noise $\epsilon_\theta(z_t,t)$ by a neural model $\epsilon_\theta$ at time step $t$, given $c$ a text prompt as condition. During training, the latent variable $z$ is obtained by passing the image into a pre-trained encoder [16]. For inference, a decoder is employed to convert the denoised latent into an image. In addition to text prompts, our model also conditions on subject representation, rendering an image generation architecture with multimodal conditions.

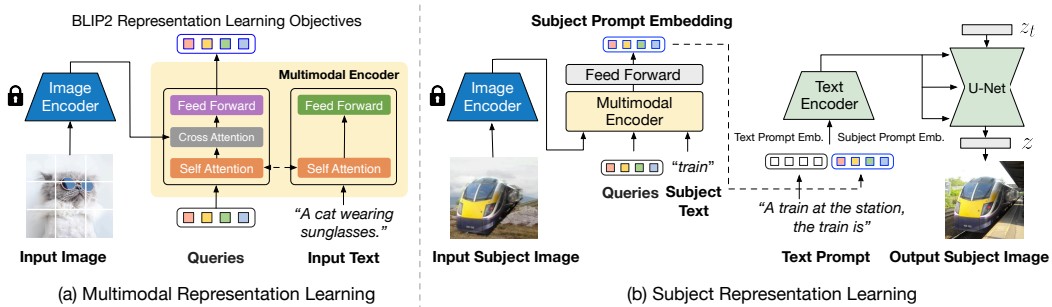

Figure 2: Illustration of the two-staged pre-training for BLIP-Diffusion. **Left**: in the multimodal representation learning stage, we follow prior work [12] and pretrain BLIP-2 encoder to obtain text-aligned image representation. **Right**: in the subject representation learning stage, we synthesize input images by composing subjects with random background. The BLIP-2 is then tasked to produce subject prompt embedding, which is later used by the latent diffusion model to generate output subject image. The image encoder remains frozen during pre-training.

## 2.2 Subject-driven Text-to-Image Generation

Given a few images of a subject, the task of subject-driven text-to-image generation aims at generating the subject in novel context based on text prompts. In the era of diffusion models, Textual Inversion [8] proposes to represent visual concepts using a placeholder text embedding, and optimize the embedding to reconstruct the subject images. DreamBooth [9] shares a similar methodology while additionally fine-tunes the diffusion model, which leads to better expressiveness and subject fidelity. One known drawback for both methods is their lengthy fine-tuning time for each new subject, which prevents the approaches from easily scaling up. More recent efforts attempt to reduce the required time and effort for fine-tuning. Concurrent to our effort, the work [11, 17, 18] pre-train the diffusion model on domain-specific images, such as cat and human face images. These models provide class-specific prior for generation thus being more efficient for fine-tuning. However, they are also constrained to a narrow list of subject categories and are not able to easily generalize to generic subjects. The work SuTI [19] proposes a knowledge distillation approach, which learns zero-shot generation from millions of fine-tuned expert models. Their model shows less flexibility in subject poses and is likely to be distracted by the background of the input images. In contrast, the pre-trained representation in our model is generic to a wide range of subjects, while generalizing efficiently to different subjects.

## 3 Method

We propose BLIP-Diffusion, the first image diffusion model that features multimodal control through built-in generic pre-trained subject representation. Specifically, we adapt BLIP-2 encoder to extract multimodal subject representation, which is later used together with text prompt to guide generation.

We aim to learn subject representation that captures the subject-specific visual appearances while in the meantime aligns well with the text prompt. To this end, we propose a two-staged pre-training strategy as shown in Figure 2. First, a multimodal representation learning stage produces text-aligned generic image representation. Second, a subject representation learning stage prompts the diffusion model with text and subject representation for subject-driven generation. In this section, we delineate the model design and pre-training strategies.

### 3.1 Multimodal Representation Learning with BLIP-2

We use Stable Diffusion [6] as the latent diffusion model, which relies on CLIP [20] text encoder to produce prompt embeddings. In order to guide the generation using both text and subject representation as prompt, it is important that the subject embedding and the text embedding are well-aligned to ensure they can cooperate with each other. Inspired by the recent vision-language pre-trained model BLIP-2 [12], which produces high-quality text-aligned visual representation, we decide to adapt it to extract text-aligned subject representation.

Specifically, as shown in Figure 2a, we employ two main modules from BLIP-2 to learn multimodal representation: a frozen pre-trained image encoder to extract generic image features, and a multimodal

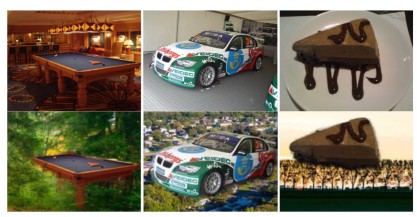 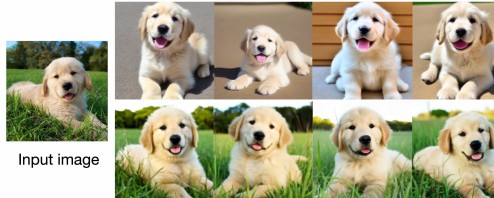

Input image

Figure 3: **Left**: example training image pairs, target images (top) and input images (bottom) with random background. Images are from the OpenImage dataset [22]. **Right**: comparison between the subject-driven image generation of our model (top), and image variations from DALLE-2 [4] (bottom). We prompt our model with the text prompt "a dog" and the subject representation, while feeding the input image into DALLE-2 for variation generation. Our model learns subject representation with minimal background information encoded, which enables faithful and flexible control when used together with text prompts.

encoder (*i.e.* Q-Former) for image-text alignment. The multimodal encoder is a transformer that accepts a fixed number of learnable query tokens and an input text. The query tokens interact with text through self-attention layers, and interact with frozen image features through cross-attention layers, and produces text-aligned image features as output. The output is of the same dimension as the number of query tokens. Empirically, we find that the originally implemented 32 output features often overpower the CLIP text embeddings when used in combination for image generation. Therefore, we instead half the number of query tokens and output 16 features.

Following BLIP-2 pre-training, we jointly train three vision-language pre-training objectives, including an image-text contrastive learning (ITC) loss that aligns the text and image representation by maximizing their mutual information, an image-grounded text generation (ITG) loss that generates texts for input images, and an image-text matching (ITM) loss that captures fine-grained image-text alignment via a binary prediction. We conduct multimodal representation learning on generic image-text paired data, which allows the model to learn a diverse set of visual and textual concepts.

## 3.2 Subject Representation Learning with Stable Diffusion

As the result of the multimodal representation learning, we obtain text-aligned visual representation of the input image. These features capture the generic semantic information of the input image. However, they are not specifically tailored to serve as guidance for the diffusion model. To this end, the subject representation learning stage aims to enable the diffusion model to leverage such visual representation, and generate different renditions of the subjects when combining with text prompts. In particular, we consider two desired properties when injecting the subject representation into a diffusion model. First, we expect the subject representation to well coordinate with the text prompts for the purpose of text-guided subject-driven generation. In this regard, prior methods [9, 18, 19] do not address the text prompts during training. They are thus not directly suitable to be used for scalable pre-training. Second, the behavior of the underlying diffusion model should ideally be maintained. This allows the subject-driven generation model to take advantage of techniques built on top of the original model on the fly, such as image editing and structure-controlled generation.

**Model Architecture.** The proposed model architecture is shown in Figure 2b. We connect the output of the BLIP-2 multimodal encoder to the input of the diffusion model's text encoder. During pre-training, the multimodal encoder takes as input a subject image and a text of the subject category, and produces a category-aware subject visual representation. We then transform the subject representation using a feed-forward layer consisting of two linear layers with GELU [21] activation in-between. The projected features are appended to the text prompt token embeddings as a soft visual subject prompt. Specifically, when combining the text token and subject embeddings, we use the template "[text prompt], the [subject text] is [subject prompt]". We pass text tokens through the CLIP embedding layer to obtain text token embeddings. We then concatenate subject embeddings and text token embeddings before passing them to the subsequent CLIP model layers. The resultant CLIP embeddings serve as guidance for the diffusion model to generate the output image. The soft visual prompt makes minimal architectural change to the underlying diffusion model, rendering an effective solution to inject subject representation while in the meantime largely inherits the modeling capabilities of the underlying diffusion model.

**Subject-generic Pre-training with Prompted Context Generation.** We aim to pre-train the model such that it learns to represent generic subjects from the input image. To this end, a naive approach

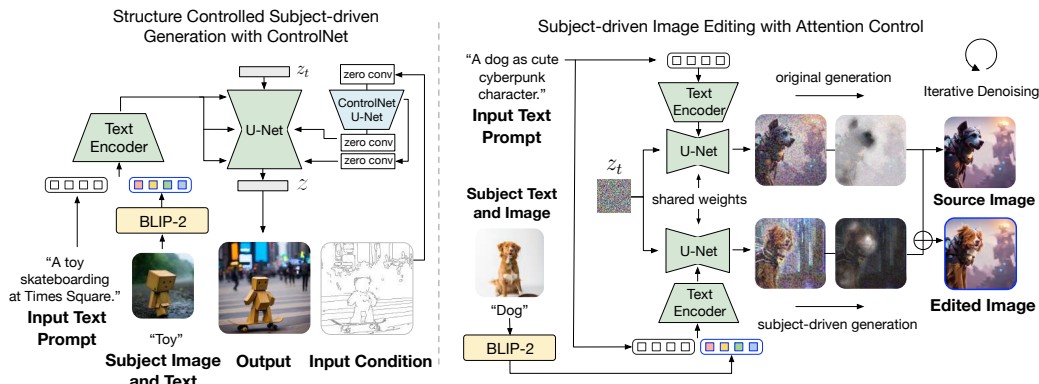

Figure 4: **Left**: BLIP-Diffusion cooperates with ControlNet for structure and subject controllable generation. Our model maintains the modeling capabilities of the underlying diffusion model and requires no re-training of the ControlNet parameters. **Right**: combined with cross-attention control techniques present in prompt-to-prompt, our model can be used for subject-driven image editing. The figure illustrates one denoising step where we mix latent maps generated with and without subject representation.

is to use the same image as both input to the multimodal encoder and output to the diffusion model as in [8, 9]. However, our preliminary experiments suggest that this leads to trivial solutions where generations are significantly interfered by the background in the inputs, or even models copying the input image as output, rendering generations not respecting the text prompts. On the other hand, while it is possible to collect multiple images of the same subject in different context, and thereby using different images as input and target, such an approach is laborious to scale up to generic subjects.

To address these issues, we propose a new pre-training task for learning subject-generic representation, called *prompted context generation*, where we curate input-target training pairs by synthesizing images of the subject in random background. The model takes the synthesized subject image as input, and aims to generate the original subject image as output according to a text prompt. Specifically, given an image containing a subject, we first feed the image and a category text of the subject to a text-prompted segmentation model CLIPSeg [23] with confidence thresholding. We then build a trimap by taking the segmentation map with higher confidence as the known foreground, the lower confidence as the uncertain region and the rest as the known background. Given the trimap, we use closed-form matting [24, 25] to extract the foreground, namely the subject. Then we compose the extracted subject onto a random background image via alpha blending. Finally, we use the synthetic image as the input and the original subject image as the output to serve as one training image pair.

As shown in Figure 3, such synthetic pairs effectively separate the foreground subject and the background context, preventing the subject-irrelevant information from being encoded in the subject prompt. In this way, we encourage the diffusion model to consider jointly the subject prompt and the text prompt for generation, leading to a pre-trained model that can be faithfully and flexibly controlled by both the subject image and the text prompt.

During pre-training, we freeze the image encoder and jointly train the BLIP-2 multimodal encoder along with the text encoder and U-Net of the latent diffusion model. To better preserve the original text-to-image generation capability, we find it beneficial to randomly drop the subject prompt at a 15% probability while using only text prompts to guide the diffusion.

### 3.3 Fine-tuning and Controllable Inference

The pre-trained subject representation enables both zero-shot generation and efficient fine-tuning for specific custom subjects. In addition, our model provides high-level visual control while inheriting the modeling capabilities of the underlying diffusion model. This enables us to leverage established image generation and editing techniques on the fly with BLIP-Diffusion as the foundation generation model. Below we first describe the efficient few-step subject-specific fine-tuning for custom subject generation. Then we present the extension capabilities of BLIP-Diffusion by incorporating existing techniques including ControlNet [1] and prompt-to-prompt image editing [2].

**Subject-specific Fine-tuning and Inference.** The pre-trained generic subject representation enables efficient fine-tuning for highly personalized subjects. Given a few subject images and the subject category text, we first use the multimodal encoder to obtain the subject representation individually. We then initialize the subject prompt embedding using the mean subject representation of all the subject images. In this way, we cache the subject prompt embedding without needing a forward pass of the multimodal encoder during fine-tuning. The diffusion model is fine-tuned to generate subject images as target by considering the text prompt embedding and the mean subject embedding. We also freeze the text encoder of the diffusion model, which we find helpful to counteract overfitting to target images. We use batch size 3 and a constant learning rate of 5e-5 with AdamW [26] optimizer across all the subjects, and generally observe decent results after 40-120 training steps, which takes 20-40 seconds to complete on a single A100 GPU.

**Structure-controlled Generation with ControlNet.** Our model introduces a multimodal conditioning mechanism for subject-control. In the meanwhile, the architecture is also compatible to integrate with ControlNet [1] to achieve simultaneous structure-controlled and subject-controlled generation. Figure 4 illustrates such integration, where we attach the U-Net of the pre-trained ControlNet to that of BLIP-Diffusion via residuals. In this way, the model takes into account the input structure condition, such as edge maps and depth maps, in addition to the subject cues. Since our model inherits the architecture of the original latent diffusion model, we observe satisfying generations using off-the-shelf integration with pre-trained ControlNet without further training.

**Subject-driven Editing with Attention Control.** Our model combines subject prompt embeddings with text prompt embeddings for multimodal controlled generation. Inspired by prompt-to-prompt [2], our model enables subject-driven image editing by manipulating the cross-attention maps of prompt tokens. In Figure 4, we show such capability where the model edits the original image with subject-specific visuals. For this purpose, we assume the generation process of the original image is known, or can be derived via inversion [13, 27] for real images. To edit the image, we first specify text tokens to edit, for example the token "dog". Next, we use the cross-attention maps of the specified token to extract automatically a mask for regions to edit. In order to preserve the layout and semantics in unedited regions, we keep the attention maps from the original generation while generating new attention maps for the inserted subject embeddings. We mix the denoising latents at each step based on the extracted editing mask. Namely, latents of the unedited regions are from the original generation whereas latents of the edited regions are from the subject-driven generation. In this way, we obtain the edited image with subject-specific visuals while also preserving the unedited regions.

## 4 Experiments

### 4.1 Pre-training Datasets and Details.

For multimodal representation learning, we follow BLIP-2 [12] and pretrain the model on 129M image-text pairs, including 115M image-text pairs from LAION [28] with CapFilt [29] captions, COCO [30], Visual Genome [31] and Conceptual Captions [32, 33]. We use ViT$_{Large}$ from CLIP [20] as the image encoder, and initialize Q-Former with BERT$_{base}$ [34]. As aforementioned, we use 16 queries to learn subject representation. Other training hyperparameters follow [29].

For subject representation learning, we use a subset of 292K images from OpenImage-V6 [22], each containing a salient subject. We also remove images with human-related subjects. We use BLIP-2 OPT$_{6.7B}$ to generate captions as text prompts. We obtain a set of 59K background images from the web to synthesize subject inputs. We use Stable Diffusion v1-5 as the foundation diffusion model. We use a total batch size 16 with a constant learning rate 2e-6 for 500K steps using AdamW [26] optimizer, taking 6 days to finish on 16 A100 40Gb GPUs. More details on hyperparameters and data filtering procedure are included in Appendix for reference.

### 4.2 Experimental Results

**Main Qualitative Results.** In Figure 5, we show qualitative generation results of BLIP-Diffusion. Thanks to the pre-trained subject representation, our model facilitates zero-shot subject-driven generation (row #1), producing meaningful results even for highly customized subjects. The model also enables efficient fine-tuning (row #3-6), demonstrating high-fidelity generation for re-contextulization, artistic stlyization, textual modification, property modification and accessorization. Compared with

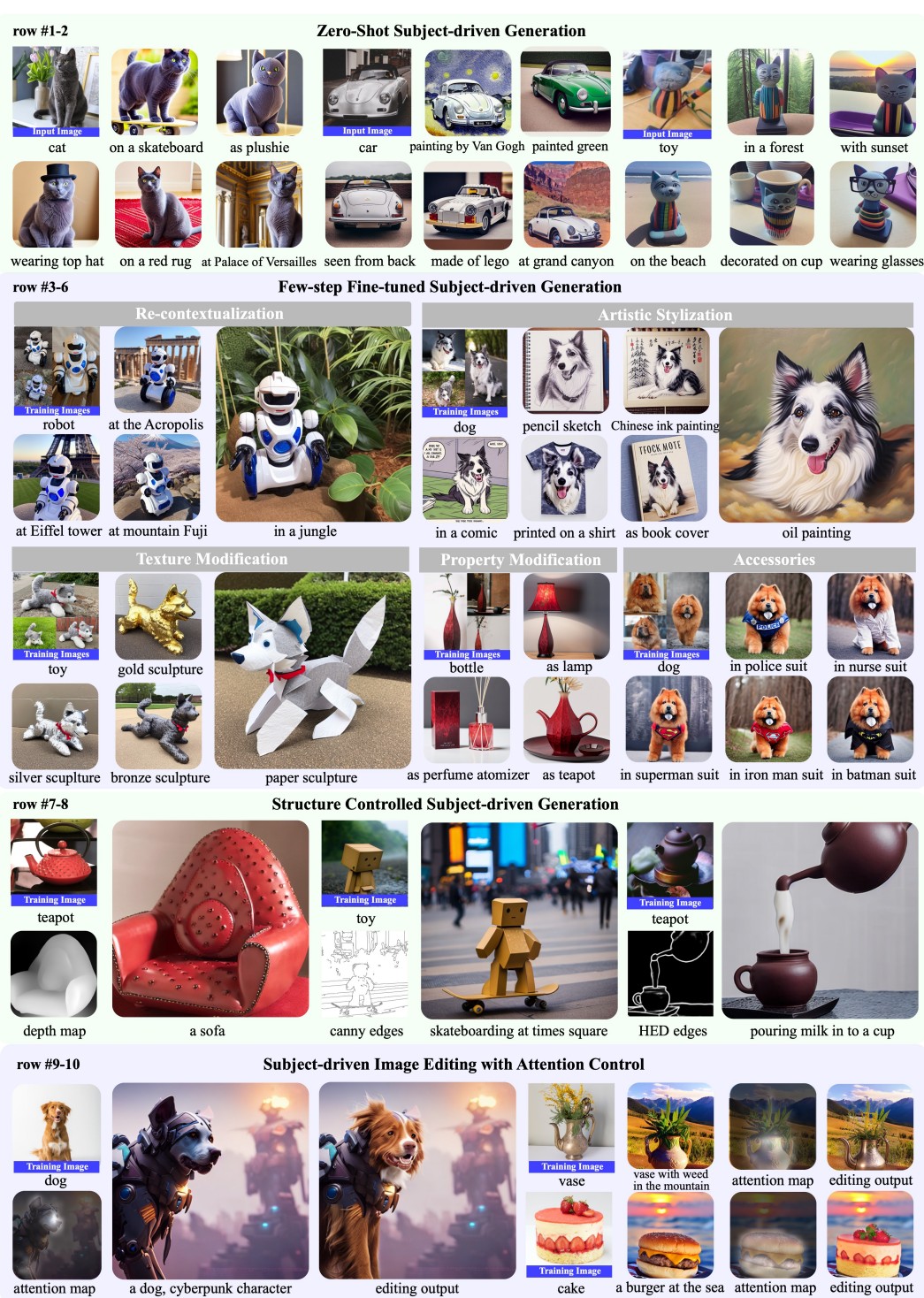

Figure 5: **Qualitative results categorized by generative capabilities**. The pre-trained subject representation enables zero-shot subject-driven generation (**row #1-2**) and efficient, high-fidelity finetuning (**row #3-6**). Our model can also serve as a foundation text-to-image generation model and leverages on the fly exsiting techniques developed on latent diffusion models. Combining with ControlNet (**row #7-8**), BLIP-Diffusion enables controllable generation using both subject and structure conditions; combining with prompt-to-prompt (**row #9-10**), our model achieves subject-driven image editing by manipulating cross-attention maps.

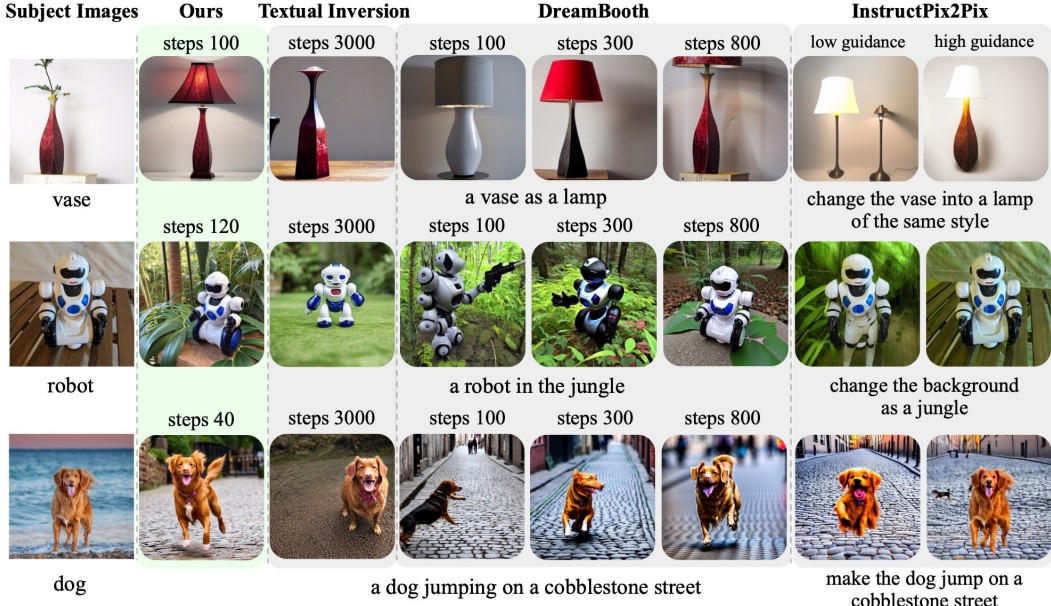

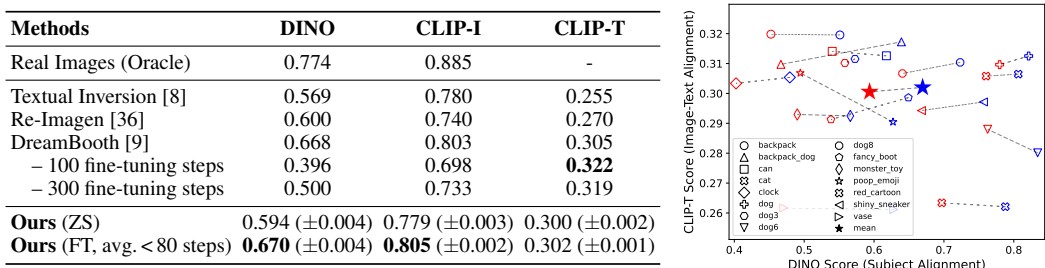

Figure 6: **Qualitative comparison** between BLIP-Diffusion with text-driven image generation methods [8, 9] and the image editing method [35]. Our model achieves high subject fidelity and prompt relevance, while needing significantly fewer finetuning steps. Example subject images from the dreambooth dataset [9] are shown, which are also used as source images for InstructPix2Pix to edit.

| Methods | DINO | CLIP-I | CLIP-T |
|---|---|---|---|
| Real Images (Oracle) | 0.774 | 0.885 | - |
| Textual Inversion [8] | 0.569 | 0.780 | 0.255 |
| Re-Imagen [36] | 0.600 | 0.740 | 0.270 |
| DreamBooth [9] | 0.668 | 0.803 | 0.305 |
| – 100 fine-tuning steps | 0.396 | 0.698 | **0.322** |
| – 300 fine-tuning steps | 0.500 | 0.733 | 0.319 |
| **Ours** (ZS) | 0.594 (±0.004) | 0.779 (±0.003) | 0.300 (±0.002) |
| **Ours** (FT, avg. < 80 steps) | **0.670** (±0.004) | **0.805** (±0.002) | 0.302 (±0.001) |

Table 1: **Left**: Quantitative comparisons on DreamBench. We report average metrics and differences across 10 experiment runs with different set of random seeds, in zero-shot (ZS) and fine-tuning (FT) setups. **Right**: Alignment metrics in zero-shot (red) and fine-tuning (blue) setups for sample subjects.

existing solutions, BLIP-Diffusion requires much less fine-tuning effort, usually 40-120 steps which are up to x20 times more efficient than previous work [8, 9]. In addition, when combining with ControlNet (row #7-8), our model can achieve simultaneous control over structure and subject. Finally, our model can introduce subject information into image editing pipeline, enable to edit images with specific subject visuals (row #9-10). These applications demonstrate potentials of using BLIP-Diffusion as a foundation text-to-image generation model with multimodal controls.

**Comparisons on DreamBooth Dataset.** We compare BLIP-Diffusion with methods [8, 9, 35, 36] (details in appendix) on DreamBooth dataset [9], containing 30 subjects, each with 4-7 images. In Figure 6, we show qualitative comparisons. Our model achieves significantly better subject fidelity than Textual Inversion, Re-Imagen and InstructPix2Pix. Compared with DreamBooth, our model exhibits comparable or better generation quality while requiring a significant lower number of fine-tuning iterations, which validates the effectiveness of our pre-trained subject representation.

In Table 1, we follow [9] and report DINO [37], CLIP-I [20] and CLIP-T scores. DINO and CLIP-I scores measure subject alignment and CLIP-T measures image-text alignment (see appendix for detailed description of the metrics). We generate 4 images for each text prompt, amounting in total 3,000 images for all the subjects. We repeat generations with 10 fixed set of random seeds and report average scores. The overall results are consistent with the qualitative findings, where BLIP-Diffusion is superior to Textual Inversion and Re-Imagen while showing comparable performance to DreamBooth while requiring less fine-tuning effort. In particular, our zero-shot generations are better

| Ablation Setups | DINO | CLIP-I | CLIP-T |
|---|---|---|---|
| BLIP-Diffusion (250K steps) | 0.566 | 0.773 | 0.299 |
| – w/o multimodal pre-training | 0.521 ↓ | 0.743 ↓ | 0.290 ↓ |
| – w/o training text encoder | 0.568 ↑ | 0.782 ↑ | 0.288 ↓ |
| – w/o subject text | 0.565 ↓ | 0.772 ↓ | 0.298 ↓ |
| – w/o subject dropping | 0.559 ↓ | 0.766 ↓ | 0.291 ↓ |

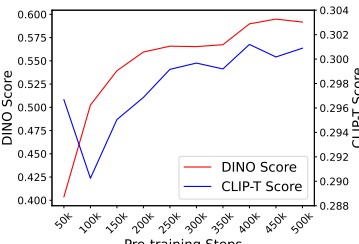

Table 2: **Left**: Ablation results. **Right**: Effect of subject representation learning with varying pre-training steps.

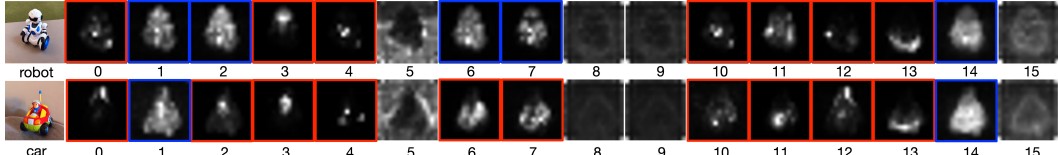

Figure 7: Attention visualization of subject representation. Leftmost is the generation using prompt "a robot/car" with subject representation injected. The learned subject embeddings capture both local (red) and holistic (blue) subject visuals. Subjects are from the DreamBench [9] dataset.

than fine-tuned Textual Inversion results. Additionally, we show per-subject metrics and observe that fine-tuning significantly improves subject alignment. In the meanwhile, fine-tuning also improves image-text alignment on average. When fine-tuning harms the image-text alignment, it is due to the model overfitting to target inputs thus resulting in generations irrespective of the text prompt. This is in particular an issue when the provided subject images are of limited visual diversity.

**Aesthetics Measures.** We use LAION aesthetic scorer to measure the visual quality of the generations. The aesthetic scorer predicts a score bounded between 0 and 10 using CLIP ViT-L/14 features, where 10 represents the highest aesthetic score. We calculate the aesthetic scores for 3,000 generations on the DreamBooth dataset and report comparison results with those produced by DreamBooth. As indicated in Table 3, results show that our model shows a clear advantage over DreamBooth models in terms of the aesthetic scores. In addition, results also demonstrate that fine-tuning is beneficial to promote high-quality generations.

**Ablation Studies.** We conduct ablation studies using 250K subject representation learning steps. Table 2 shows zero-shot evaluation results. Our findings are: (i) it is critical to conduct multimodal representation learning (Section 3.1), which bridges the representation gap between subject embeddings and text prompt embeddings. (ii) freezing text encoder of the diffusion model worsens the interaction between subject embedding and text embedding. This leads to generations copying subject inputs and not respecting the text prompts. Despite leading to higher subject alignment scores, it does not allow text control, falsifying the task of text-to-image generation. (iii) Giving subject text to the multimodal encoder is helpful to inject class-specific visual priors, thereby leading to moderate improvement in metrics. (iv) Pre-training with random subject embedding dropping helps to better preserve the diffusion model's generation ability, thus benefiting the results. We further demonstrate the effect of subject representation learning. The figure (right) shows that both image-text alignment and subject alignment improve with growing pre-training steps of subject representation learning.

**Subject Representation Visualization.** It is observed that pixels are attracted more to embeddings that describe them [2]. Following this observation, in Figure 7, we visualize the learned subject embeddings using cross-attention maps. The figure shows that the learned embeddings encode

| Models | Aesthetic Scores |
|---|---|
| BLIP-Diffusion (fine-tuned) | 6.50 |
| BLIP-Diffusion (zero-shot) | 6.43 |
| DreamBooth | 6.20 |

Table 3: Aesthetic scores comparison between BLIP-Diffusion models and DreamBooth models. Our models show a quantitative advantage over DreamBooth models.

fine-grained yet different aspects of the subject. For example, certain embeddings (*e.g.* 0, 3, 4, 10-13) tend to focus on more local features while others (*e.g.* 1, 14) encode more holistic visuals. This demonstrates the complimentary effect of employing multiple subject embeddings.

**Limitations.** Our model suffers from common failures of subject-driven generation models, such as incorrect context synthesis, overfitting to training set as detailed in [9]. In addition, it inherits some weakness of the underlying diffusion model, which may fail to understand text prompts and fine-grained composition relations. We show some of such failure examples in Figure 8. Despite the limitations, the proposed technique is generic to harvest future development of diffusion models.

## 5  Conclusion

This paper proposed BLIP-Diffusion, a new text-to-image diffusion model with built-in multimodal control capabilities powered by BLIP-2 [12]. The model is pre-trained using a two-stage strategy to learn progressively multimodal subject representation, which facilitates high-fidelity zero-shot and efficient fine-tuned subject-driven generation. BLIP-Diffusion produces better zero-shot generations than fine-tuned models such as Textual Inversion. It also achieves up to 20x fine-tuning speed up than best prior methods with comparable generation quality. In addition, it can work in conjunction with other established techniques, such as ControlNet and prompt-to-prompt, for image generation and editing with simultaneous structure and subject control. We consider BLIP-Diffusion as an important step towards building foundational text-to-image generation model with multimodal control.

## Acknowledgement

We thank colleagues at Salesforce AI Research for support and discussions, and anonymous program committee members for their voluntary reviews and valuable feedback.

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

# A    Appendix

## A.1    Broader Impact

Image generation models are susceptible to be used as tools for generating false content or prompting misinformation. Subject-driven generation could be misused as a tool for generating fake image of individuals. To mitigate this issue, our model has been trained on generic objects where person-related subjects have been purposely removed from the training data. This makes the model weaker at generating fake images using person as subject control.

Our model is built using the pre-trained Stable Diffusion model trained on web-scraped datasets. Therefore, our model inherits some of its shortcomings, such as generating biased contents with social stereotypes, or other NSFW contents if used inappropriately. Our model's ability to precisely control the generation subject can help mitigate certain biases. We can use NSFW detectors to block potential inappropriate content from being generated. Nevertheless, we strongly caution against using our model directly in user-facing applications without a careful inspection of the model's output. Proper content moderation and regulation are highly advised to prevent undesirable consequence.

## A.2    Failure Cases Analysis

In Figure 8, we outline common failure cases of the model. Our model suffers from issues observed for prior subject-driven generation models as outlined in [9], including incorrect context synthesis, overfitting to training set. In addition, it subsumes some weakness of the underlying diffusion model, such as failing to address text prompts or generating fine-grained composition relations.

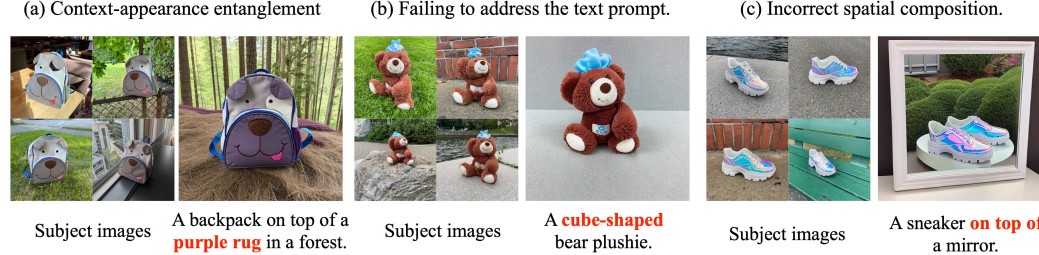

(a) Context-appearance entanglement (b) Failing to address the text prompt. (c) Incorrect spatial composition.

Subject images | A backpack on top of a **purple rug** in a forest. | Subject images | A **cube-shaped** bear plushie. | Subject images | A sneaker **on top of** a mirror.

Figure 8: Example failure generations. Subject images used for finetuning are shown on the left.

## A.3 Competing Methods

We compare BLIP-Diffusion with fine-tuning based [8, 9] and retrieval-augmented [36] subject-driven generation models on the public DreamBench dataset [9]. We also compare qualitatively with the image editing method InstructPix2Pix [35]. We briefly introduce these methods below.

- *Textual Inversion* [8]: a fine-tuning method which optimizes a placeholder embedding to reconstruct the training set of subject images. It requires 3,000 training steps for learning a new concept, which takes around 30 minutes on an A100 GPU.

- *DreamBooth* [9]: a fine-tuning method similar to textual inversion. In addition to the placeholder embedding, it also optimizes parameters of the U-Net for a total budget of around 800 steps. We report intermediate results using 100 and 300 fine-tuning steps, while refer to metrics reported by the authors for full model comparison. Fine-tuning DreamBooth on a new concept costs around 6 minutes on an A100 GPU.

- *Re-Imagen*: a retrieval-augmented model, which takes the subject images as references and attend to them to generate new images. While the model requires no tuning, it significantly underperforms other models. The model is not publicly available, thus we do not have access to qualitative examples for comparison.

- *InstructPix2Pix*: an image editing model, which takes as input the source image and an editing instruction to generate edited images. Although it does not represent explicitly subjects, it can be used for applications such as subject re-contextualization and property modification. Therefore, we also include it for qualitative comparison. In particular, we experiment with both low (1.0) and high (1.5) image guidance scales, where a low image guidance scale preserves less the subject while promotes the text alignment; a high image guidance scale preserves better the original image yet is more likely to overlook the editing instruction.

## A.4 Evaluation Metrics

We adopt metrics proposed in DreamBooth [9] for evaluation, including DINO, CLIP-I and CLIP-T scores. Among them, DINO and CLIP-I scores are used to measure subject fidelity and CLIP-T is used to measure image-text alignment. DINO score is the average pairwise cosine similaity between the ViT-S/16 DINO embeddings of the generated and real images. CLIP-I score is the average pairwise CLIP ViT-B/32 image embeddings of the generated and real images. It is considered that DINO score is the preferred metric for measuring subject fidelity as it is sensitive to the differences between subjects of the same class. CLIP-T score is the average cosine similarity between prompt and image CLIP embeddings.

To better evaluate and compare subject-driven text-to-image models, it is suggested that these metrics should be considered jointly to avoid biased conclusion. For example, a model that naively copies the training set images will produce high DINO and CLIP-I scores with low CLIP-T scores. In the other case, a vanilla text-to-image generation model without subject knowledge, *e.g.* stable diffusion, will produce high CLIP-T scores with poor subject alignment. Both models are not considered desirable for the subject-driven text-to-image generation task.

### A.5 Pre-training Datasets

For multimodal representation learning, we use the same pre-training data as by BLIP-2, totaling 129M images. This includes COCO [30], Visual Genome [31], CC3M [32], CC12M [33], SBU [38] and 115M images from LAION400M [28]. We also employ the synthetic captions created using CapFilt method [29] for web images. We refer interested readers to Section 3.4 in the BLIP-2 paper [12] for details of the data bootstrapping configurations.

For subject representation learning, we use a subset of OpenImage-V6. We filter the data using the annotations provided by the dataset. In particular, we discard a sample if it satisfies one of the following cases: (i) a group of objects of the same class appear in the image; (ii) the image is taken from inside of the subject; (iii) the object is of aspect ratios larger than 2; (iv) objects occupy a too large (0.8) or too small (0.3) area relative to the image; (v) human-related subject, including boy, girl, person, man, mammal, woman, human body, human head, human hair, human arm, human face, human leg, human hand, human foot, human eye, human mouth, human nose, human ear, clothing, suit; (vi) cluttered objects, including tree, plant, houseplant, desk, table, poster and billboard. This results in 292K images for subject representation learning.

### A.6 Fine-tuning, Inference and Evaluation on DreamBooth Dataset

For all fine-tuning experiments, we use AdamW [26] optimizer with constant learning rate 5e-6 and no warm-up steps. We use batch size 3, adam beta1 0.9, adam beta2 0.999, adam epsilon 1e-8 and weight decay 0.01. We fine-tune models on a single A100 (40Gb) GPU and select checkpoints manually based on a set of validation prompts. We report the number of iterations for each subject on DreamBench below, on average 76 steps, taking around 40 seconds to complete on a single A100.

For inference, we use PNDM scheduler [39] for 100 denoising steps. We use a fixed guidance scale 7.5 for all experiments.

Table 4: Number of fine-tuning steps for DreamBench subjects.

| | | | | | |
|---|---|---|---|---|---|
| backpack | 110 | backpack-dog | 110 | bear-plushie | 110 |
| bowl | 40 | can | 70 | candle | 80 |
| cat | 40 | cat2 | 50 | clock | 120 |
| colorful-sneaker | 80 | dog | 50 | dog2 | 50 |
| dog3 | 40 | dog5 | 20 | dog6 | 40 |
| dog7 | 50 | dog8 | 40 | duck-toy | 60 |
| fancy-boot | 50 | grey-sloth-plushie | 70 | monster-toy | 120 |
| pink-sunglasses | 90 | poop-emoji | 90 | rc-car | 120 |
| red-cartoon | 70 | robot-toy | 110 | shiny-sneaker | 80 |
| teapot | 120 | vase | 120 | wolf-plushie | 80 |

In Table 5 and 6, we report average metrics across 10 experiment runs for each subject in the dataset, in zero-shot and fine-tuning setups, respectively.

### A.7 Zero-shot Subject-driven Image Manipulation

Our model is able to extract subject features to guide the generation. In addition to applications of subject-driven generations and editing, we show that such pre-trained subject representation enables intriguing and useful applications of zero-shot image manipulation, including subject interpolation and subject-driven style transfer.

*Subject Interpolation.* It is also possible to blend two subject representation to generate subjects with a hybrid appearance. This can be achieved by traversing the embedding trajectory between subjects. In Figure 9, we create bilinear interpolations among 4 different subject representations, and render the interpolated subject in a novel context. As the figure shows, the subject appearance blends along the trajectory and fits naturally with the environment. This is useful when multiple subjects are used as reference to guide the generation. For example, subject interpolation can be used in joint with subject-driven style transfer to create hybrid style from multiple guiding subjects.

Table 5: Average metrics for each subject on DreamBench in zero-shot setup.

| Subject | backpack | backpack-dog | bear-plushie | berry-bowl | can | candle |
|---|---|---|---|---|---|---|
| **DINO** | 0.452 | 0.467 | 0.634 | 0.750 | 0.540 | 0.395 |
| **CLIP-I** | 0.782 | 0.712 | 0.739 | 0.792 | 0.641 | 0.710 |
| **CLIP-T** | 0.320 | 0.310 | 0.304 | 0.257 | 0.314 | 0.316 |

| Subject | cat | cat2 | clock | colorful-sneaker | dog | dog2 |
|---|---|---|---|---|---|---|
| **DINO** | 0.760 | 0.703 | 0.402 | 0.680 | 0.780 | 0.730 |
| **CLIP-I** | 0.835 | 0.854 | 0.735 | 0.769 | 0.849 | 0.831 |
| **CLIP-T** | 0.306 | 0.286 | 0.303 | 0.298 | 0.310 | 0.307 |

| Subject | dog3 | dog5 | dog6 | dog7 | dog8 | duck-toy |
|---|---|---|---|---|---|---|
| **DINO** | 0.558 | 0.705 | 0.763 | 0.656 | 0.641 | 0.665 |
| **CLIP-I** | 0.747 | 0.788 | 0.867 | 0.817 | 0.816 | 0.840 |
| **CLIP-T** | 0.310 | 0.313 | 0.288 | 0.309 | 0.307 | 0.287 |

| Subject | fancy-boot | grey-sloth-plushie | monster-toy | pink-sunglasses | poop-emoji | rc-car |
|---|---|---|---|---|---|---|
| **DINO** | 0.538 | 0.632 | 0.490 | 0.599 | 0.494 | 0.569 |
| **CLIP-I** | 0.800 | 0.755 | 0.734 | 0.836 | 0.689 | 0.761 |
| **CLIP-T** | 0.291 | 0.315 | 0.293 | 0.308 | 0.307 | 0.281 |

| Subject | red-cartoon | robot-toy | shiny-sneaker | teapot | vase | wolf-plushie |
|---|---|---|---|---|---|---|
| **DINO** | 0.697 | 0.534 | 0.668 | 0.451 | 0.471 | 0.463 |
| **CLIP-I** | 0.826 | 0.787 | 0.759 | 0.804 | 0.786 | 0.737 |
| **CLIP-T** | 0.263 | 0.315 | 0.294 | 0.314 | 0.262 | 0.327 |

*Subject-driven Style Transfer.* When provided with a subject, the model can encode the appearance style of it and transfer to other subjects. We refer such an application as subject-driven style transfer. In Figure 10 and 11, we generate stylized reference subjects with the aid of edge-guided ControlNet. The styles are hinted by the guiding subjects. Specifically, we feed BLIP-2 with guiding subjects and their category texts, *e.g.* fire, flower, glass, vase, ball, bread, to extract the subject representation. In this application, guiding subjects serve as alternative of textual prompts to specify styles. This is useful especially when a style is non-trivial to describe by natural languages accurately.

### A.8 Additional Qualitative Results and Subject Fidelity Showcasing

In Figure 12 to 14, we provide additional qualitative results on DreamBench subjects and prompts. We show the reference subject image in the first column. In the rest columns, we provide generated renditions. To showcase subject fidelity and photorealism, we purposely mix one genuine subject image in and leave for interested readers to figure out. Read the captions to verify.

Table 6: Average metrics for each subject on DreamBench in fine-tuning setup.

| Subject | backpack | backpack-dog | bear-plushie | berry-bowl | can | candle |
|---------|----------|--------------|--------------|------------|-----|--------|
| **DINO** | 0.551 | 0.639 | 0.693 | 0.808 | 0.618 | 0.519 |
| **CLIP-I** | 0.839 | 0.760 | 0.752 | 0.829 | 0.695 | 0.752 |
| **CLIP-T** | 0.320 | 0.317 | 0.307 | 0.254 | 0.313 | 0.311 |

| Subject | cat | cat2 | clock | colorful-sneaker | dog | dog2 |
|---------|-----|------|-------|------------------|-----|------|
| **DINO** | 0.806 | 0.747 | 0.479 | 0.739 | 0.821 | 0.793 |
| **CLIP-I** | 0.869 | 0.864 | 0.784 | 0.805 | 0.860 | 0.841 |
| **CLIP-T** | 0.306 | 0.284 | 0.305 | 0.320 | 0.313 | 0.307 |

| Subject | dog3 | dog5 | dog6 | dog7 | dog8 | duck-toy |
|---------|------|------|------|------|------|----------|
| **DINO** | 0.573 | 0.727 | 0.834 | 0.672 | 0.723 | 0.699 |
| **CLIP-I** | 0.751 | 0.801 | 0.891 | 0.823 | 0.823 | 0.838 |
| **CLIP-T** | 0.312 | 0.311 | 0.280 | 0.310 | 0.310 | 0.284 |

| Subject | fancy-boot | grey-sloth-plushie | monster-toy | pink-sunglasses | poop-emoji | rc-car |
|---------|------------|--------------------|-------------|-----------------|------------|--------|
| **DINO** | 0.649 | 0.717 | 0.566 | 0.625 | 0.627 | 0.651 |
| **CLIP-I** | 0.827 | 0.780 | 0.743 | 0.826 | 0.784 | 0.775 |
| **CLIP-T** | 0.299 | 0.322 | 0.292 | 0.312 | 0.290 | 0.288 |

| Subject | red-cartoon | robot-toy | shiny-sneaker | teapot | vase | wolf-plushie |
|---------|-------------|-----------|---------------|--------|------|--------------|
| **DINO** | 0.788 | 0.626 | 0.757 | 0.484 | 0.628 | 0.599 |
| **CLIP-I** | 0.882 | 0.803 | 0.804 | 0.819 | 0.812 | 0.760 |
| **CLIP-T** | 0.262 | 0.316 | 0.297 | 0.331 | 0.261 | 0.325 |

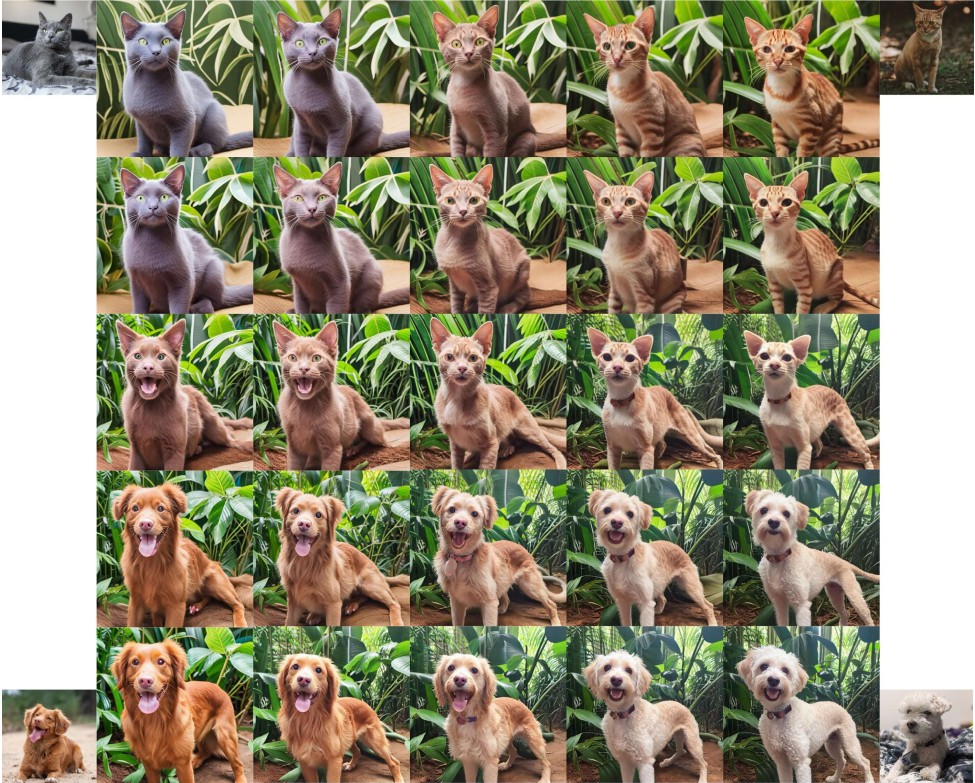

Figure 9: Zero-shot subject interpolation. We interpolate subject representation and use the same denoising and decoder network for generation. The intermediate subject representation naturally blends the subject appearance, while fitting coherently into the new context.

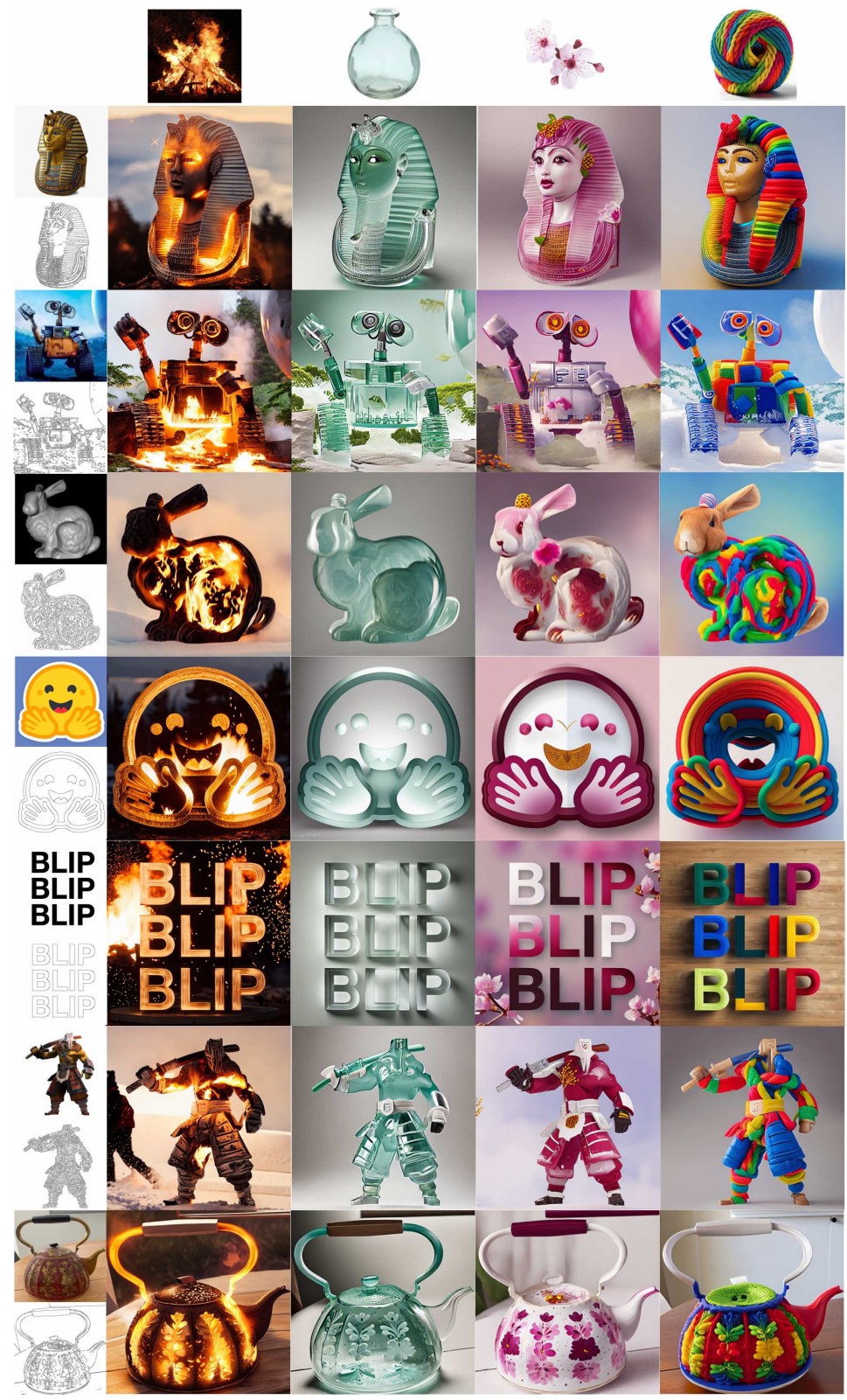

Figure 10: Zero-shot subject-driven stylization. We show guiding subject images on top. In the rest rows, we show reference subjects and their canny maps on left, and stylized reference subjects by column.

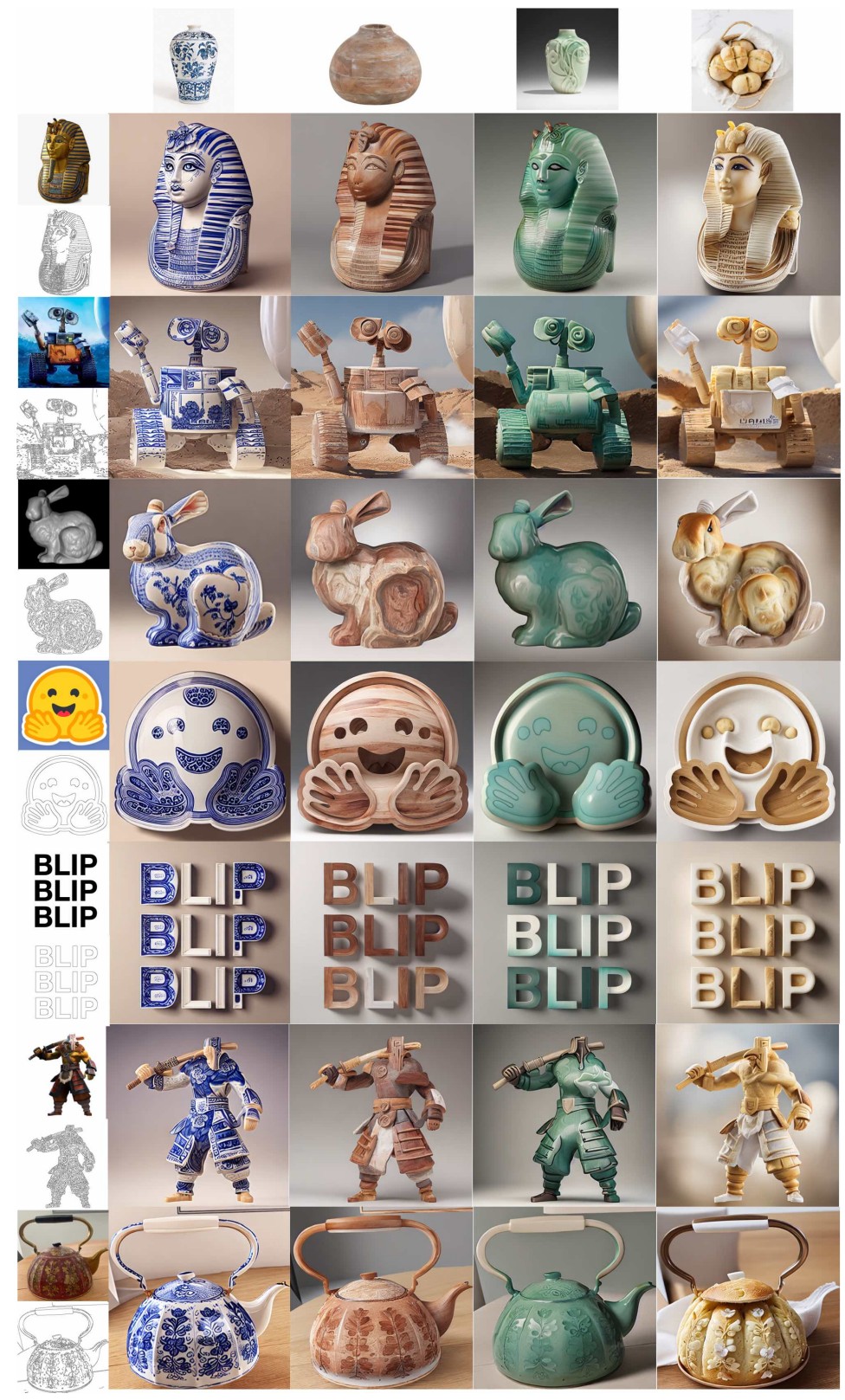

Figure 11: (Cont.) Zero-shot subject-driven stylization. We show guiding subject images on top. In the rest rows, we show reference subjects and their canny maps on left, and stylized reference subjects by column.

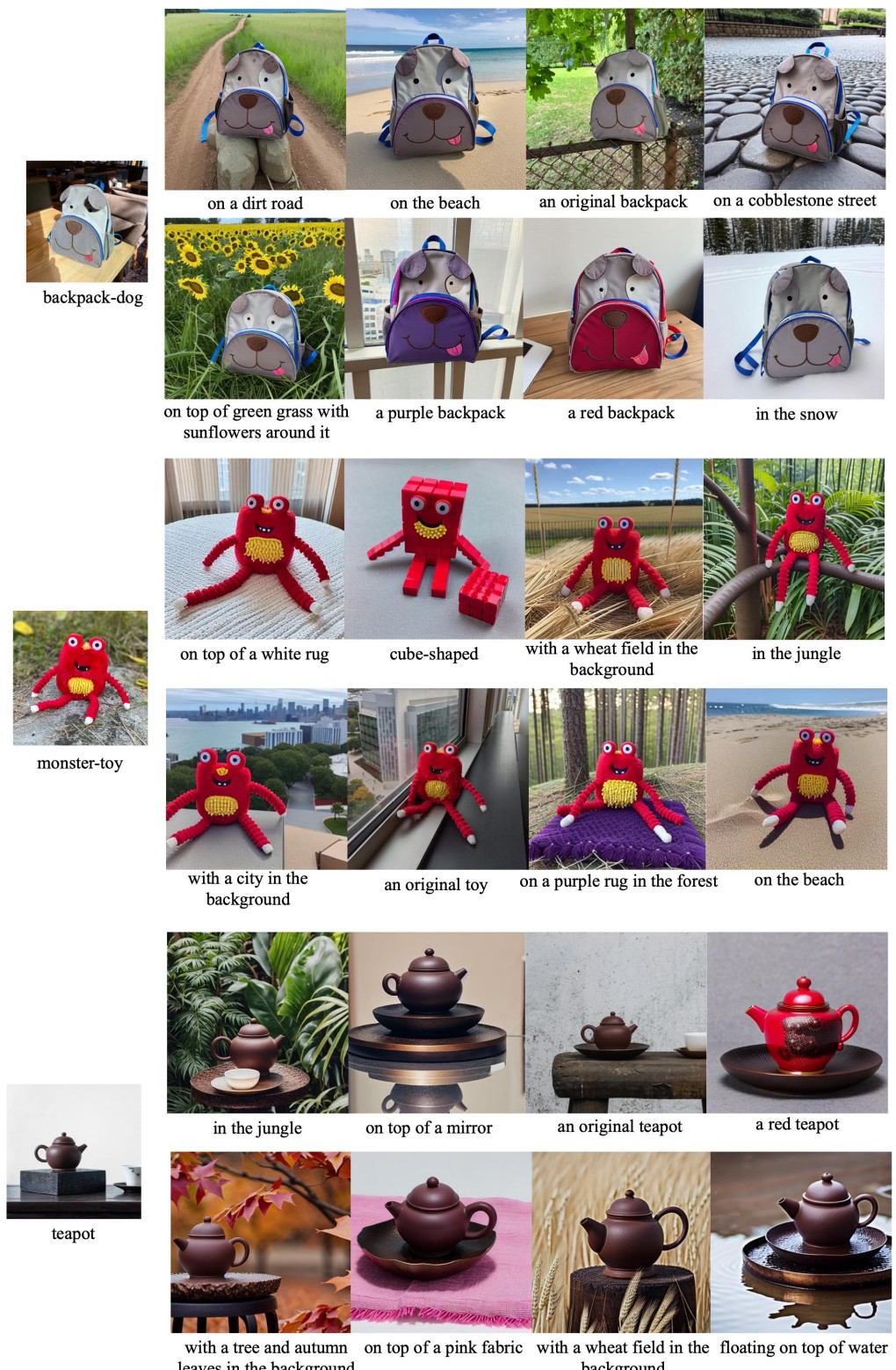

Figure 12: Additional qualitative results using DreamBench subjects and prompts. To showcase subject fidelity and photorealism, we mix one genuine subject image in the generations for readers to figure out. Zoom-in and read the captions to verify.

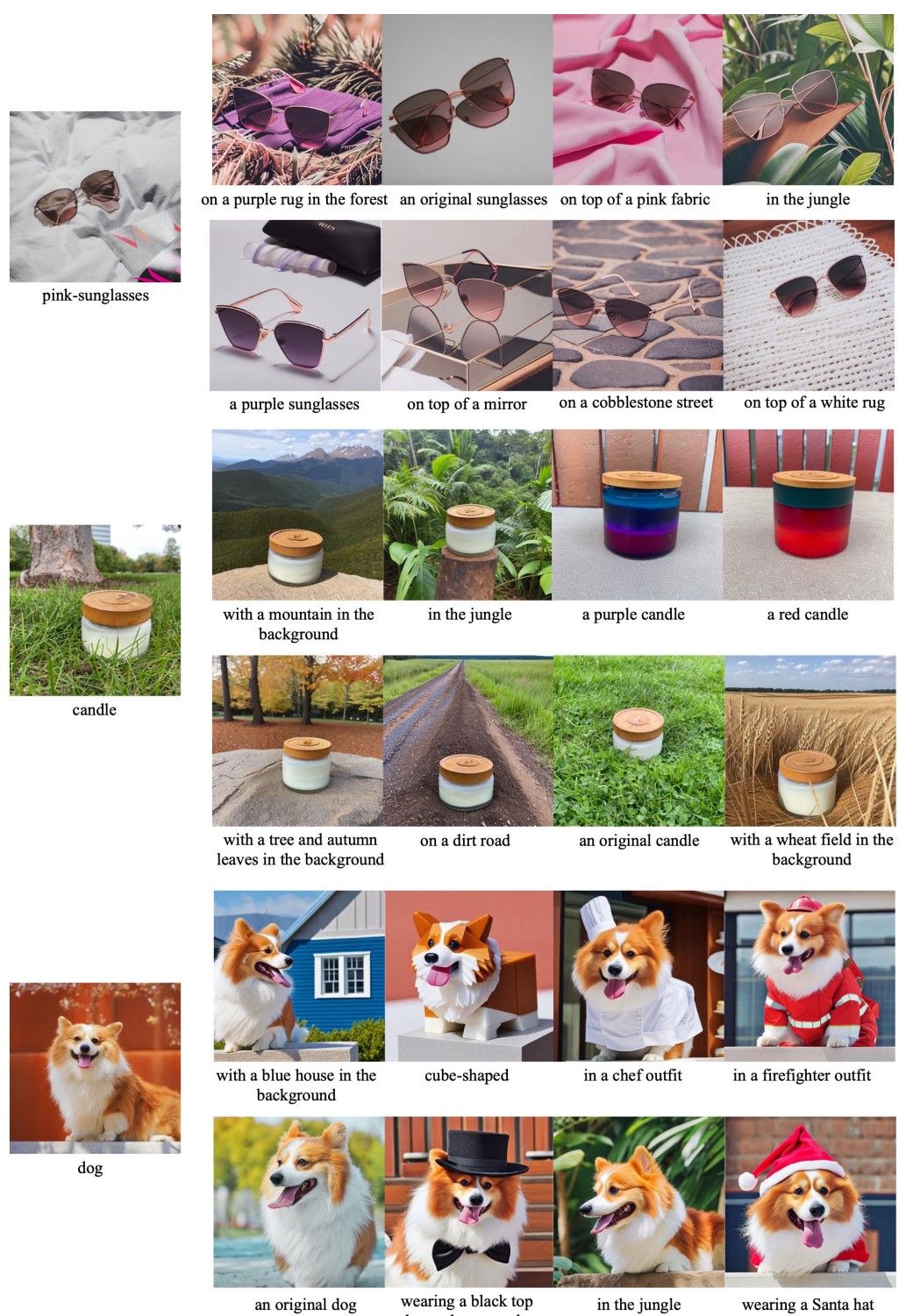

Figure 13: (Cont.) Additional qualitative results using DreamBench subjects and prompts. To showcase subject fidelity and photorealism, we mix one genuine subject image in the generations for readers to figure out. Zoom-in and read the captions to verify.

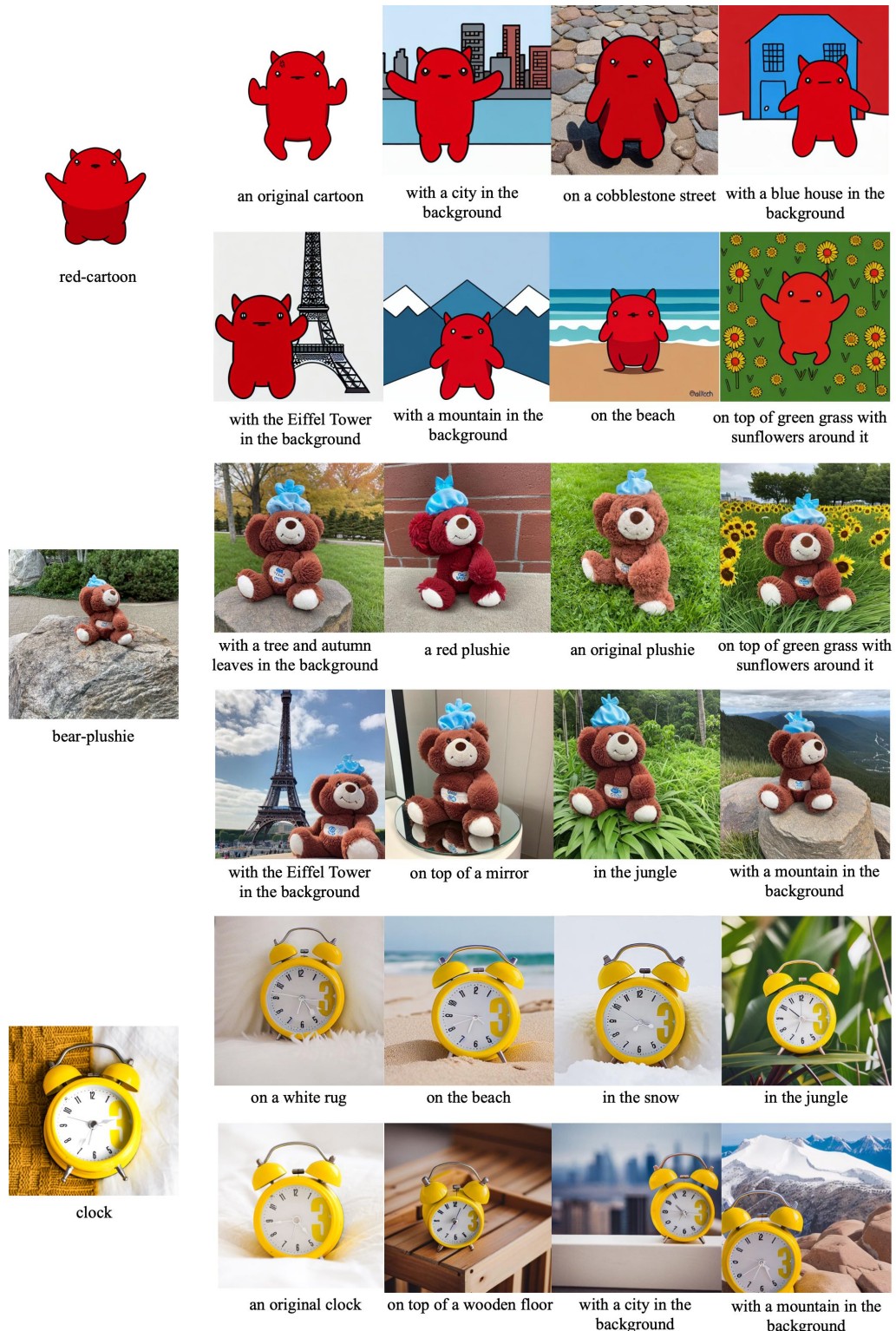

red-cartoon

an original cartoon

with a city in the background

on a cobblestone street

with a blue house in the background

with the Eiffel Tower in the background

with a mountain in the background

on the beach

on top of green grass with sunflowers around it

bear-plushie

with a tree and autumn leaves in the background

a red plushie

an original plushie

on top of green grass with sunflowers around it

with the Eiffel Tower in the background

on top of a mirror

in the jungle

with a mountain in the background

clock

on a white rug

on the beach

in the snow

in the jungle

an original clock

on top of a wooden floor

with a city in the background

with a mountain in the background

Figure 14: (Cont.) Additional qualitative results using DreamBench subjects and prompts. To showcase subject fidelity and photorealism, we mix one genuine subject image in the generations for readers to figure out. Zoom-in and read the captions to verify.

