# OpenReview forum: "BLIP-Diffusion: Pre-trained Subject Representation for Controllable Text-to-Image Generation and Editing"
_NeurIPS.cc/2023/Conference — NeurIPS 2023 poster_

### Official Review · Reviewer_wCQb · 2023-07-03

**Soundness:** 3 good
**Presentation:** 4 excellent
**Contribution:** 2 fair
**Rating:** 6
**Confidence:** 4

**Summary:**

This paper proposes  BLIP-Diffusion, a new subject-driven image generation model with multimodal encoder that supports multimodal control which consumes inputs of subject images and text prompts.

**Strengths:**

The proposed method is novel and enables subject-driven generation under efficient fine-tuning and zero-shot setups.

The proposed model can serve as a foundation model and combined with previous methods such as prompt-to-prompt and controlnet.

Solid experimental results and ablations.

**Weaknesses:**

The contribution is a bit lacking. It seems to me that the work is just a combination and the multimodal encoder from CLIP and the t2i model.

The improvement under efficient fine-tuning setup seems to be incremental.

**Questions:**

I am curious about the contribution of the proposed pretraining approach. How's the performance for the naive pretraining strategy (keep the same image as both input to the multimodal encoder and output to the diffusion mode.)?

How do you get the random background image for generating synthetic image?

For finetuning compared with Dreambooth, does it take the same time for each timestep?

What's the inference time cost of the proposed method compared with standard stable diffusion?

**Limitations:**

Yes.

---

> ### Author Rebuttal · Authors · 2023-08-07
>
> We thank the reviewer for confirming the novelty of our approach. We address questions below.
>
> -------
> **Q1**: contribution of the work?
>
> **A1**:  We summarize the scientific significance of BLIP-Diffusion as below:
> - **BLIP-Diffusion represents a novel approach to subject-driven generation using multimodal encoder**. Previous work (DreamBooth, Textual Inversion) learns subject embeddings via inversion. Our approach using multimodal encoders represents a novel and generic technique that has proved more efficient than inversion. In addition, our approach can also benefit from the advancement of multimodal vision-language foundation models, offering better potentials for stronger subject-driven generative capabilities;
> - **BLIP-Diffusion highlights a new two-staged pre-training strategy for category-generic subject-driven generation**. The multimodal representation learning stage harvests the high-quality text-aligned visual features. The subject representation learning stage includes a novel pre-training task prompted context generation, ensuring the subject visuals and text prompt can well coordinate for generation. Both stages are category-generic and require no domain-specific annotations, which make BLIP-Diffusion stand out from concurrent work.
> - **Zero-shot subject-driven generative capabilities are unprecedented**. Zero-shot generation with highly-customized and category-generic subjects is a challenging task. Such zero-shot capabilities were not available in prior models. We enable this novel capability via the newly introduced subject representation learning stage, which represents a significant advancement in this domain per se.
> BLIP-Diffusion features a foundational architecture that enables versatile applications. Different from existing work, our model’s generative capabilities are showcased in multiple applications, including generation, editing, geometry-guided generation, image manipulation/stylization (see supplementary) and subject interpolation (see supplementary). This demonstrates the flexibility of our model and its potential to serve as a foundation subject-driven generation model.
> - **BLIP-Diffusion demonstrates preferable generation results while offering significant speed-up in finetuning**. Specifically, our model fine-tunes 20x more efficiently than DreamBooth. This effectively reduces fine-tuning time per subject from 5-10 minutes (500-1000 fine-tuning steps) to sub-minute (50-100 fine-tuning steps). This has important implications on applications where fine-tuning efficiency matters, such as multimodal dialogues.
> - We provide quantitative evaluation results on public datasets with category-generic subjects, which validate effectiveness of the model. Our model will be open-sourced for researchers and practitioners for reproducing our results and findings.
>
> --------
> **Q2**: Performance if using the same image for multimodal encoder and target.
>
> **A2**: As described in Ln 142, this setup leads to trivial solutions where the image is directly copied. In our experiments, the resultant model always reproduces the input image, failing to address text prompts.
>
> This observation also echoes with the findings as reported by DreamBooth, where the authors propose additional regularization methods to counteract the issue. Their proposed regularization technique, however, does not easily scale up for scalable pre-training.
> In this regard, with input background replaced, we effectively avoid such optimization shortcuts and enforce the model to learn to condition on both subject visuals and text prompt for the generation.
>
> ----
>
> **Q3**: How do you get the random background images?
>
> **A3**: We download 59K images from the royalty-free photo stock website by querying for background and landscape images, as these images usually contain less salient or distracting subjects.
>
> ----
> **Q4**: For finetuning compared with DreamBooth, does it take the same time for each timestep?
>
> **A4**: **Yes, our model takes the same time for each iteration as DreamBooth**.
> As described in section 3.3, we only fine-tune U-Net to specialize for custom subjects. Subject embeddings are pre-computed at the cost of merely one forward pass with negligible time needed, thus no multimodal encoder is needed during fine-tuning.
> To get a better idea of the wall-clock time cost, on DreamBooth Benchmark, DreamBooth models on average take 6-10 minutes and fine-tune for 600-1000 steps, while our model costs less than a minute for around 80 steps on average.
>
> ------
> **Q5**: What's the inference time cost of the proposed method compared with standard stable diffusion?
>
> **A5**: **Inference of BLIP-Diffusion is as efficient as standard stable diffusion models**.
> For zero-shot inference, only one additional pass of multimodal encoder is needed to initialize the subject embedding. This cost is negligible compared to the iterative denoising step.
> For inference with fine-tuned model checkpoints, subject embeddings are loaded as part of the model. No additional inference cost is needed.
>
> ------
> **Q6**: Improvement under efficient fine-tuning seems incremental.
>
> **A6**: Fine-tuning aims to learn better subject visuals for highly customized subjects. This consequently leads to better subject alignment scores, i.e. DINO, CLIP-I as indicated in Table 1. Such results demonstrate the effectiveness of fine-tuning.
> In contrast, our fine-tuning setup does not explicitly optimize for prompt understanding, thus showing no significant effect on CLIP-T score, which is naturally expected. While our work does not emphasize on the design of fine-tuning procedure, it may be also possible to integrate better calibrated fine-tuning and inference methods (e.g. [1]).
>
> We will revise the manuscript to highlight this discussion.
>
> [1] Key-Locked Rank One Editing for Text-to-Image Personalization, Tewel, Yoad and Gal, Rinon and Chechik, Gal and Atzmon, Yuval, SIGGRAPH 2023.
>
> -------
> Hope the response addresses the questions.

---

> > ### Comment · Reviewer_wCQb · 2023-08-17
> >
> > Thanks a lot for authors' detailed responses. My concerns are mostly addressed. But I am still not quit sure about the efficiency improvement. As said, the "Subject embeddings are pre-computed" and the model takes the same time for each iteration. It is unclear to me why the model takes much less steps to fine-tune. Is the comparison fair? Are you using the same backbone as dreambooth? Now I keep my original score.

---

> > > ### Author Response · Authors · 2023-08-17
> > >
> > > Thanks for your question and please kindly find our response below.
> > >
> > > As the title highlights, the improved fine-tuning efficiency comes from the “pre-trained subject representation”. This was motivated by the inefficiency of DreamBooth as described in LN 27, which we reiterate here:
> > >
> > > “*We attribute such inefficiency to the fact that most pre-trained text-to-image models do not natively support multimodal control - using both images and texts as control input. As a result, it becomes challenging to learn subject representation that aligns with the text space while capturing the subject visuals with high fidelity*”.
> > >
> > > Technically, DreamBooth always initializes fine-tuning **from a randomly initialized embedding**. In contrast, our model intializes fine-tuning **from subject embeddings produced by multimodal encoder**, which already capture largely the subject visuals even before the fine-tuning (evidenced by the zero-shot results). This results in a much smaller visual gap to learn during fine-tuning, which translates to less fine-tuning steps required.
> > >
> > > And yes, we use Stable Diffusion as described in LN 217. The DreamBooth model we compared to are with the same backbone.
> > >
> > > Definitely please let us know if more clarification is required and we are more than happy to further address remaining concerns.
> > >
> > > Thanks.

---

> > > > ### Author Response · Authors · 2023-08-18
> > > >
> > > > Dear reviewer wCQb, please kindly let us know whether the response addresses your concerns. Your feedback is greatly valued.
> > > >
> > > > Thanks.

---

### Official Review · Reviewer_tkfj · 2023-07-03

**Soundness:** 4 excellent
**Presentation:** 4 excellent
**Contribution:** 3 good
**Rating:** 8
**Confidence:** 5

**Summary:**

This paper aims to solve the subject-driven text-to-image generation with a pre-trained subject representation that is derived from a vision-language encoder, the BLIP2 model. The obtained subject representation captures rich information of the visual input while being aligned with the textual space. The text-conditioned diffusion model is pre-trained to produce images based on this subject representation and the text input. The derived diffusion model can produce novel impressive renditions of a given subject with respect to different text prompts, in both zero-shot and few-step finetuned scenarios. Beyond that, the paper presents fancy applications like controlled generations and subject-driven stylization. Compared to the state-of-the-art approaches, the proposed BLIP-Diffusion demonstrates preferable subject-driven generation results both qualitatively and quantitatively while offering significant speed-up.

**Strengths:**

The paper explicitly leverages the text-aligned image feature and the diffusion model trained on this subject representation is able to generate subject renditions with compelling quality. The novel combination with the BLIP2 model forms an effective solution to subject-driven text-to-image generation. The method presented is neat and elegant.

Moreover, I like many technical details authors propose to achieve the final quality.  For example, authors carefully design a synthetic procedure to form the synthetic training pairs to tackle the tendency to produce trivial solutions. Also, it is found that randomly dropping the subject prompt with some probability is beneficial to text-to-image generation.

The paper demonstrates fancy applications in various applications and presents impressive generation results. the significant speed-up for the subject-driven generation makes it promising in practical usage. The quantitative study further proves the superiority of the method over prior leading approaches.








**Weaknesses:**

- One possible way to explain the method is that this paper uses a detailed text prompt instead of a rough description like "an image of [V]" as used in Dreambooth. That is, the features captured from the BLIP2 may not be the key. One simpler baseline is "Encoder-based Domain Tuning for Fast Personalization of Text-to-Image Models". The paper should mention such baseline and add more discussions regarding this.

- To avoid the distraction of the background, the authors propose to replace the background of the target image with random background. Then why not just remove the background during training?

- One primary usage of subject-driven text-to-image models is to generate images related to humans. However, the paper avoids this scenario by explicitly removing human-related images during training, why is that? It is suggested to have a discussion regarding this, otherwise, it is suspicious that the method does not perform well on portraits.

- While the paper qualitatively measures the fidelity to the input image and the text prompt respectively, there is no measure for image quality. While in the chosen samples of Figure 6 the BLIP-Diffusion shows apparent quality advantage over prior arts, it still needs a quantitative measure to reflect the image quality on a large amount of image results.T

**Questions:**

It is suggested to provide more technical details for the style transfer part. Does it need some special prompt design when specifying styles according to the input image?

**Limitations:**

The limitation is sufficiently discussed in the supplementary material.

---

> ### Author Rebuttal · Authors · 2023-08-07
>
> We thank the reviewer for confirming the technical depth and empirical values of our work. In the following, we provide response to answer reviewer's questions.
>
> ------
> **Q1**:  paper uses a detailed text prompt instead of a rough description like "an image of [V]" as used in Dreambooth. the features captured from the BLIP2 may not be the key; discussion on E4T.
>
> **A1**: We use BLIP-2 captions for pre-training. While BLIP-2 captions are more detailed than template prompts “an image of [V]”, they are not sufficient to describe the exact subject visuals. Therefore, having subject embeddings is crucial in our model.
>
> To provide evidence for this claim, we kindly refer the reviewer to Table 2, which shows that the generation performance improves with better subject embeddings. This highlights the importance of subject embedding on the generation quality. Qualitatively, in Figure 7, we show that subject embeddings learn meaningful global and local subject visuals. This further supports the effectiveness of subject embeddings.
>
> We are aware of the E4T work and have also referenced it in LN 74. It is worth highlighting that BLIP-Diffusion is category-generic, and can be applied to any customized subjects. This is supported by our quantitative evaluation on the DreamBooth datasets. In contrast, E4T pre-trains the model on domain-specific datasets, e.g. cats, thus cannot generalize easily to out-of-domain subjects.
>
> We are happy to include an expanded discussion on E4T in the revised version.
>
> -----
> **Q2**: Why not just remove the background during training?
>
> **A2**: We thank the reviewer for this insightful question.
> This will impose additional constraints on the input subject image to also have an empty background during fine-tuning and inference. As a result, it would most likely require additional procedures for background removal and introduce external reliance, which we consider suboptimal. In contrast, in our setup, the model is able to automatically identify the subject from the input image after pre-training, even with distracting background scenes.
>
> -------
> **Q3**: Application on portraits.
>
> **A3**: We thank the reviewer for raising this issue. While we acknowledge the practical values of portrait generation, we purposely avoid human-related generation for training and evaluation. This is mainly to abide by the code of ethics of the conference (https://neurips.cc/public/EthicsGuidelines), and was also motivated by related NeurIPS work [1]. With our due respect, we humbly request the reviewer to kindly share our concern.
>
> As such, we would expect the model checkpoint along this submission to not perform well for human portraits. However, we note that the proposed two-staged pre-training strategy is category-generic. Interested readers may opt to resume pre-training on domain-specific datasets to tailor for their own application scenarios.
>
> [1] PASS: An ImageNet replacement for self-supervised pretraining without humans, Yuki M. Asano, Christian Rupprecht, Andrew Zisserman, Andrea Vedaldi
>
> ------
> **Q4**: Quantitative image quality measure.
>
> **A4**: We thank the reviewer for the suggestion. We recognize that measuring image quality quantitatively for subject-driven text-to-image generation is an open research question. And to the best of us, few prior work presents relevant results.
> As such, we propose to use the aesthetic scorer (https://github.com/LAION-AI/aesthetic-predictor) to measure the visual quality of the generated images, which was used to select high-quality image data used for training state-of-the-art diffusion models.
> The aesthetic scorer predicts an aesthetic score bounded from 0 to 10 using CLIP ViT-L/14 features, where 10 is the highest aesthetic score. To get a better understanding of the score, we reference the description from LAION project page (https://laion.ai/blog/laion-aesthetics/) that out of the LAION 5-billion images:
>
> - 1.2B (24%) images have scores 4.5+;
> - 12M (0.24%) images have scores 6+;
> - 3M (0.06%) images have scores 6.25+;
> - 625K (0.01%) images have scores 6.5+;
>
> We calculate the aesthetic scores for 3000 images of our model and compare with those generated from DreamBooth. We use prompts and subjects from the DreamBooth datasets. The results are shown as below.
>
> | Setups      | Aesthetic Scores |
> | ----------- | ----------- |
> | BLIP-Diffusion (fine-tuned)    | 6.50       |
> | BLIP-Diffusion (zero-shot)   | 6.43        |
> | DreamBooth   | 6.20       |
>
> **These results show that**:
> - BLIP-Diffusion produces better image quality than DreamBooth. We attribute this to the two-staged pre-training strategy which helps to better align subject embeddings and text embeddings;
> - BLIP-Diffusion after fine-tuning produces top 0.01% quality images when compared with the 5B internet image corpora; while DreamBooth quality is among top 0.24%. This shows a clear quality advantage of our model.
> - Fine-tuning helps produce higher quality images for our model;
>
> We will include this result into the supplementary material during revision.
>
> ------
>
> We thank the reviewer for appreciating our work and hope the response addresses the questions.

---

### Official Review · Reviewer_ZD8x · 2023-07-07

**Soundness:** 2 fair
**Presentation:** 3 good
**Contribution:** 3 good
**Rating:** 5
**Confidence:** 4

**Summary:**

The paper introduces "BLIP-Diffusion", a new subject-driven image generation model that supports multimodal control using subject images and text prompts. The model introduces a pre-trained multimodal encoder to provide subject representation and enables zero-shot subject-driven generation and efficient fine-tuning for customized subjects. This model can be combined with existing techniques to enable novel subject-driven generation and editing applications.

**Strengths:**

1. Originality: The paper introduces a novel subject-driven image generation model, BLIP-Diffusion, which supports multimodal control using subject images and text prompts. This model is original in its approach as it combines a pre-trained multimodal encoder for subject representation, enabling zero-shot subject-driven generation and efficient fine-tuning for customized subjects.

2. Quality: The quality of the paper is evident in the detailed explanation of the model and the comprehensive experiments conducted to validate its performance. The paper includes qualitative results that demonstrate the model's capabilities, such as zero-shot subject-driven generation and high-fidelity fine-tuning. The model also shows high subject fidelity and prompt relevance, requiring significantly fewer fine-tuning steps compared to other methods.

3. Clarity: The paper is well-structured and clear in its presentation. The authors provide a thorough explanation of the model, its implementation, and the experiments conducted. The use of figures and tables further enhances the clarity of the paper, providing visual representations of the model's performance and capabilities.

4. Significance: The significance of the paper lies in its contribution to the field of image generation. The BLIP-Diffusion model presents a new approach to subject-driven image generation, offering potential for novel subject-driven generation and editing applications. The model's ability to perform zero-shot subject-driven generation and efficient fine-tuning for customized subjects is a significant advancement in this field.

**Weaknesses:**

Limited Zero-Shot Performance and Dependence on Fine-Tuning: The paper claims that the proposed BLIP-Diffusion model can perform zero-shot rendering of images across various categories of subjects. However, the results presented do not fully substantiate this claim. Both qualitatively and quantitatively, the zero-shot results are not as impressive as one might expect. Furthermore, the model's performance seems to heavily rely on fine-tuning. While fine-tuning is a common practice in machine learning, the extent to which the model depends on it raises questions about its practicality and efficiency. The necessity of fine-tuning to achieve good results could be seen as a limitation, especially in scenarios where rapid or on-the-fly generation is required. This dependence on fine-tuning could limit the model's applicability and ease of use in certain contexts.

**Questions:**

1. Clarification on Zero-Shot Performance: The paper claims that the BLIP-Diffusion model can perform zero-shot rendering of images across various categories of subjects. However, the results presented do not fully substantiate this claim. Could the authors provide more evidence or examples to support this claim?

2. Editability Issue: The paper suggests that using trained background replaced subject images can address the editability issue. However, it seems that this approach might only deal with recontextualization, not subject area editing. Could the authors justify this approach and explain how it addresses the editability issue?


**Limitations:**

The paper's main claim is that the proposed BLIP-Diffusion model can perform zero-shot rendering of images across various categories of subjects. However, the results presented do not fully substantiate this claim. Both qualitatively and quantitatively, the zero-shot results are not as impressive as one might expect. Furthermore, the model's performance seems to heavily rely on fine-tuning. While fine-tuning is a common practice in machine learning, the extent to which the model depends on it raises questions about its practicality and efficiency. The necessity of fine-tuning to achieve good results could be seen as a limitation, especially in scenarios where rapid or on-the-fly generation is required. This dependence on fine-tuning could limit the model's applicability and ease of use in certain contexts.

---

> ### Author Rebuttal · Authors · 2023-08-07
>
> We thank the reviewer for confirming the originality, quality, clarity and significance of our work. We provide response to reviewer's question below.
>
> -----
> **Q1**: Zero-shot performance and its applicability.
>
> **A1**: As described in Section 3, BLIP-Diffusion is the very first model to unlock the zero-shot subject-driven generation capabilities for generic categories. The significance in its zero-shot capability is summarized below:
>
> 1. **Zero-shot subject-driven generative capabilities are unprecedented**. Zero-shot generation with highly-customized and category-generic subjects is a challenging task. Such zero-shot capabilities were not available in prior models. We enable this novel capability via the newly introduced subject representation learning stage, which represents a significant advancement in this domain per se.
>
> 2. **Zero-shot subject-driven generation quality outperforms some recent fine-tuning methods**. Quantitatively, Table 1 shows that the zero-shot result outperforms significantly the recent fine-tuning based model Textual Inversion (TI). In particular, BLIP-Diffusion achieves zero-shot DINO score 0.594 (versus 0.569 for TI), and CLIP-T score 0.300 (versus 0.255 for TI). This demonstrates the competitive performance of the zero-shot generation with the prior art.
>
> 3. **Whether fine-tuning is needed depends on the application**. Apart from subject-driven generation, BLIP-Diffusion also facilitates various additional zero-shot applications. We kindly refer the reviewer to the supplementary material Figure 3, Figure 4 for more examples, where we show results using BLIP-Diffusion for stylization / image manipulation. Specifically, our model can use the appearance style of an input subject to guide the generation. In this application, despite no fine-tuning being performed, our model produces visually appealing and creative generations. In light of this, we consider that fine-tuning is not an absolute dependency, and its necessity can be determined based on the application.
>
> While being able to generate and edit in the zero-shot fashion is an appealing capability, it is important to underscore the novel concept of pre-trained subject representation and the two-staged pre-training strategy.  In this regard, we consider the zero-shot capabilities of BLIP-Diffusion as compelling evidence of the effectiveness of our pre-training approach.
>
> -----
> **Q2**: Fine-tuning limits the ease of use in certain contexts.
>
> **A2**: This is precisely the motivation of introducing pre-trained subject representation into the text-to-image generation model. With such pre-trained representation, BLIP-Diffusion significantly improves the subject-driven fine-tuning efficiency by up to 20 times, compared to the state-of-the-art model DreamBooth. This effectively reduces fine-tuning time per subject from 5-10 minutes (500-1000 fine-tuning steps) to sub-minute (50-100 fine-tuning steps). Whilst potential further improvement is possible, we consider BLIP-Diffusion clearly superior in terms of fine-tuning efficiency than leading solutions.
>
> -----
> **Q3**: Justification of the prompted context generation pre-training task with background replacement.
>
> **A3**: We appreciate the reviewer's question on how background replacement helps subject representation learning. The referred “edibility” requires (1) transferring of subject visuals; (2) guided generation according to text prompts. Our background replacement fulfills both requirements. Even though the pre-training only recontextualizes the subject, the model can generalize to other text prompts and directly edit the subject.  More technically, the prompted context generation task learns a joint subject-text space, which allows any text prompt to interact with the subject visuals.
>
> -----
> We hope the response addresses reviewer's questions. We will revise the manuscript accordingly for better clarity.

---

> > ### Author Response · Authors · 2023-08-15
> >
> > Hi reviewer, we appreciate your time and effort in providing reviews and we have provided rebuttal accordingly.
> >
> > Does the rebuttal address your concerns? Please kindly let us know for remaining feedback.

---

### Official Review · Reviewer_dpqw · 2023-07-07

**Soundness:** 3 good
**Presentation:** 3 good
**Contribution:** 3 good
**Rating:** 6
**Confidence:** 5

**Summary:**

The paper addresses the issues of lengthy fine-tuning and preserving the subject fidelity in subject-driven text-to-image generation models. Different from existing models such as Textual Inversion and Dreambooth that invert subject visuals into text embedding space, the paper introduces a new multimodal encoder which is pre-trained to produce visual representation aligned with the text. New subject renditions are then generated using such visual representation. The paper highlights the utility of the proposed approach in zero-shot subject-driven generation and editing applications, and demonstrates 20x speedup in fine-tuning for customized subjects.

**Strengths:**

The paper is well-motivated and well-written. The key idea of the work to deeply align subject embedding and the text embedding is interesting. The proposed model is compatible with ControlNet and Prompt-to-Prompt, which has potential to unleash several important editing capabilities. The experiment results are solid, with sufficient evaluations.

**Weaknesses:**

I see the key novelty of the proposed approach in section 3.2 where the paper introduces the subject representation learning. The paper mentions that output of the BLIP-2 multimodal encoder is passed to CLIP Text Encoder by combining the text and subject embeddings. I suggest authors elaborate this to provide more concrete details. We get embeddings as output from CLIP Text Encoder right? How is it possible to combine them before passing as input to CLIP Text Encoder. I might be missing something here, would be great if the paper clarifies this in rebuttal. Also, it is not clear if the paper additionally finetunes the CLIP Text encoder. In lines 164-164, it is mentioned that the Text encoder is also fine-tuned. But wouldn’t that bring language drift issues? Are there any specific strategies used by the paper in fine-tuning text encoder that aid in avoiding the language drift problem? Please clarify. Also, in section 3.2, how many synthetic pairs are used? I am happy to revise my final rating based on the clarifications in the rebuttal.

**Questions:**

Please see weaknesses.

**Limitations:**

Authors adequately addressed the limitations.

---

> ### Author Rebuttal · Authors · 2023-08-07
>
> We thank the reviewer for confirming the novelty of our approach and the comprehensive results. We address reviewer's question as below.
>
> -----
> **Q1**: “We get embeddings as output from CLIP Text Encoder right? How is it possible to combine them before passing as input to CLIP Text Encoder ”
>
> **A1**: Thanks for the question. As described in Section 3.2, we use subject embeddings from BLIP encoder as soft visual prompts. Specifically, given a text prompt,
> we first pass text tokens through the CLIP embedding layer, thus obtaining text token embeddings;
> we then concatenate subject embeddings and the text token embeddings;
> the combined embeddings are passed to subsequent CLIP layers, i.e. the positional embedding layer, then the self-attention layers.
> We will revise LN131-136 to make this more clarified.
>
> -----
> **Q2**: “It is mentioned that the text encoder is also fine-tuned. But wouldn’t that bring language drift issues?”
>
> **A2**: As referred to by DreamBooth paper, language drift causes the model to “associate the class name with the specific instance”, due to fine-tuning the CLIP text encoder which overfits to the subject appearance.
>
> Different from DreamBooth, our model uses a subject embedding produced by a new BLIP encoder. The CLIP text encoder is tasked to align the subject embedding with the text prompt, rather than learning the subject appearance. Therefore, we reduce the risk of the text encoder overfitting to specific subject instances. Furthermore, we give captions to the text encoder during training to preserve its language capability.
>
> Empirically, we do not observe the issue of language drift. As a qualitative evidence, in the subject editing application (Figure 5, #9-10), our model generates images well-aligned with the text either with subject embeddings (images after editing), or with text embeddings only (images before editing). This is also evidenced quantitatively by Table 1, which shows that our model produces comparable or better text alignment than prior work.
>
> -----
> **Q3**: “How many synthetic pairs are used”?
>
> **A3**: We use a subset of 292K images from OpenImage-V6, as described in Section 214, Ln-214.	In the meanwhile, please kindly find the detailed description of the selection criterion in the supplementary material A.5. We are happy to expand the description in the revised version for better clarity.
>
> -------
>
> **Below, please allow us to reiterate the significance of our work**.
>
> - **BLIP-Diffusion represents a novel approach to subject-driven generation using multimodal encoder**. Previous work (DreamBooth, Textual Inversion) learns subject embeddings via inversion. Our approach using multimodal encoders represents a novel and generic technique that has proved more efficient than inversion. In addition, our approach can also benefit from the advancement of multimodal vision-language foundation models, offering better potentials for stronger subject-driven generative capabilities;
> - **BLIP-Diffusion highlights a new two-staged pre-training strategy for category-generic subject-driven generation**. The multimodal representation learning stage harvests the high-quality text-aligned visual features. The subject representation learning stage includes a novel pre-training task prompted context generation, ensuring the subject visuals and text prompt can well coordinate for generation. Both stages are category-generic and require no domain-specific annotations, which make BLIP-Diffusion stand out from concurrent work.
> - **Zero-shot subject-driven generative capabilities are unprecedented**. Zero-shot generation with highly-customized and category-generic subjects is a challenging task. Such zero-shot capabilities were not available in prior models. We enable this novel capability via the newly introduced subject representation learning stage, which represents a significant advancement in this domain per se.
> BLIP-Diffusion features a foundational architecture that enables versatile applications. Different from existing work, our model’s generative capabilities are showcased in multiple applications, including generation, editing, geometry-guided generation, image manipulation/stylization (see supplementary) and subject interpolation (see supplementary). This demonstrates the flexibility of our model and its potential to serve as a foundation subject-driven generation model.
> - **BLIP-Diffusion demonstrates preferable generation results while offering significant speed-up in finetuning**. Specifically, our model fine-tunes 20x more efficiently than DreamBooth. This effectively reduces fine-tuning time per subject from 5-10 minutes (500-1000 fine-tuning steps) to sub-minute (50-100 fine-tuning steps). This has important implications on applications where fine-tuning efficiency matters, such as multimodal dialogues.
> We provide quantitative evaluation results on public datasets with category-generic subjects, which validate effectiveness of the model. Our model will be open-sourced for researchers and practitioners for reproducing our results and findings.
>
> -----
> **Q4**: I am happy to revise my final rating based on the clarifications in the rebuttal.
>
> **A4**: We hope the response clarifies the questions. We humbly request the reviewer to re-evaluate the significance of our work based on the response above.

---

> > ### Author Response · Authors · 2023-08-15
> >
> > Hi reviewer, we appreciate your time and effort in providing reviews and we have provided rebuttal accordingly.
> >
> > Does the rebuttal address your concerns? Please kindly let us know for remaining feedback.

---

> ### Comment · Reviewer_dpqw · 2023-08-15
>
> Thanks for the detailed responses to my questions on combining text and subject embeddings, language drift issues, and clarifying the number of synthetic pairs used. I updated my reviews.

---

### Author Response · Authors · 2023-08-14
**Welcome further response and discussion**

We appreciate the valuable reviews we receive, which help significantly to improve the manuscript.

We have provided response to reviewers' questions and are committed to address remaining concerns.

Please kindly let us know if there is additional feedback and suggestions.

---

### Decision · Program_Chairs · 2023-09-21

**Decision:**

Accept (poster)

**Comment:**

All the reviewers recommend the acceptance of the work. Reviewers appreciated the proposed multi-modal encoder strategy for subject-driven generation. Reviewers raised several clarification questions and also raised concerns regarding the quality of the zero-shot generation results. Authors addressed several of these questions in their responses. The reviewers did raise some valuable concerns and suggestions that should be addressed in the final camera-ready version of the paper, which include adding the relevant rebuttal discussions and revisions in the main paper. The authors are encouraged to make the necessary changes to the best of their ability.